

**Characterization and source apportionment of aerosol light**
**scattering in a typical polluted city in Yangtze River Delta,**
**China**
Dong Chen[1, 2], Yu Zhao[1, 3*], Jie Zhang[2, 3], Huan Yu[4], Xingna Yu[4]
1. State Key Laboratory of Pollution Control & Resource Reuse and School of the
Environment, Nanjing University, 163 Xianlin Ave., Nanjing, Jiangsu 210023, China
2. Jiangsu Provincial Academy of Environmental Science, 176 North Jiangdong Rd.,
Nanjing, Jiangsu 210036, China
3. Collaborative Innovation Center of Atmospheric Environment and Equipment
Technology, CICAEET, Nanjing, Jiangsu 210044, China
4. School of Environmental Science and Engineering, Nanjing University of
Information Science and Technology, Nanjing, Jiangsu 210044, China
*Corresponding author: Yu Zhao
Phone: 86-25-89680650; email: yuzhao@nju.edu.cn





**ABSTRACT**
Through online observation and offline chemistry analysis of samples at
suburban, urban and industrial sites (NJU, PAES and NUIST respectively) in Nanjing,
a typical polluted city in Yangtze River Delta, we optimized the aerosol light
scattering estimation method, identified its influencing factors, and quantified the
contributions of emission sources to aerosol scattering. The daily average
concentration of $PM_{2.5}$ during the sampling period (November 2015-March 2017) was
163.1 ± 13.6 μg/m$^3$ for the heavily polluted period, 3.8 and 1.6 times those for the
clean (47.9 ± 15.8 μg/m$^3$) and lightly polluted (102.1 ± 16.4 μg/m$^3$) periods,
respectively. The largest increase in PM concentration and its major chemical
components was found at the size range of 0.56-1.0 μm for the heavily polluted period,
and the contributions of nitrate and sulfate were the greatest in the 0.56-1.0 μm
fraction (19.4-39.7% and 18.1-34.7% respectively) for all the three periods. The
results indicated that the large growth of nitrate and sulfate were one of the major
reasons for the polluted periods. Based on measurements at the three sites, the US
IMPROVE algorithm was optimized to evaluate aerosol scattering in eastern China.
The light-absorption capacity OC was estimated to account for over half of the
methanol soluble organic carbon (MSOC) at NJU and PAES, whereas the fraction was
lower at NUIST. Based on Mie theory, we found that the high relative humidity (RH)
could largely enhance the light scattering effect of accumulation particles, but it had
few effects on the mixing state of particles. The scattering coefficients of particles
within the 0.56-1.0 μm range contributed the most to the total scattering (28-69%).
The mass scattering efficiency (MSE) of sulfate and nitrate increased with the
elevated pollution level, whereas a low MSE of organic matter (OM) was found for
the heavily polluted period, probably because a proportion of OM had only
light-absorption property. A coupled model of positive matrix factorization (PMF) and
Mie theory was developed and applied for the source apportionment of aerosol light
scattering. Coal burning, industry and vehicles were identified as the major sources of
the reduced visibility in Nanjing, with an estimated collective contribution at 64-70%.





The comparison between the clean and polluted period suggested that the increased
primary particle emissions from vehicles and industry were the major causes of the
visibility degradation in urban and industrial regions, respectively. In addition,
secondary aerosols were a great contributor to the reduced visibility.

## 1 INTRODUCTION

Atmospheric aerosols play a great role in visibility degradation, radiative balance
variation and climate (Liu et al., 2017; Malm and Hand, 2007; Zhang et al., 2017),
resulting largely from their light extinction (Seinfeld and Pandis, 2006).
Understanding the contributions of individual chemical species to aerosol light
extinction is important for policy making to alleviate the reduced visibility in cities
with aerosol pollution. Studies have estimated that the aerosol single scattering albedo
(the fraction of light scattering coefficient to the total extinction) ranges from
0.81-0.93 in urban China (Andreae et al., 2008; Cao et al., 2012; Xu et al., 2002; Xu
et al., 2012), implying that the deteriorated visibility primarily results from the
scattering effect of aerosols.
Aerosol light scattering is greatly affected by its chemical composition and
hygroscopic growth (Liu et al., 2008; Tao et al., 2014a). Based on estimation of the
mass scattering efficiency (MSE) of different chemical components, previous studies
found that nitrate, sulfate, sea salt and organic matter (OM) are the dominant
contributors to aerosol scattering. Developed based on the long-term observations in
national parks, the US "IMPROVE" (Interagency Monitoring of Protected Visual
Environments) algorithm has been applied to calculate the light extinction of chemical
species in aerosols (Watson et al., 2002). Two versions of IMPROVE algorithms
(IMPROVE1999 and IMPROVE2007 hereinafter) were deduced successively
(Lowenthal and Naresh, 2003; Pitchford et al., 2007), and both assumed that OM has
no light-absorption capacity and only light-scattering capacity. As part of OM,
however, brown carbon (BrC) has been highlighted in recent studies for its light
absorption in the near UV region (Alexander et al., 2008; Bond et al., 2006;
Ramanathan et al., 2007; Zhang et al., 2017), and consideration of the light-absorption



effect of OM in the optimization process of the IMPROVE formula could improve the
understanding of aerosol optical capacity by chemical species (Yan et al., 2014). In
addition, hygroscopic growth is a key factor influencing aerosol light scattering
(Schwartz, 1996). Previous studies have shown that the light scattering of sulfate and
nitrate in $PM_{2.5}$ could be largely enhanced at high relative humidity (RH) conditions
(Titos et al., 2016). Aerosol hygroscopicity is expected to depend largely on the
particle size and the abundance of water-soluble chemical components (Swietlicki et
al., 2008; Tang, 1996). Through the theoretical calculation, Liu et al. (2014) found
that smaller particles were in highly hygroscopic mode, whereas larger particles were
in nearly hydrophobic mode.

Recently, many studies have been conducted on the relationships between

visibility and aerosol light scattering in China (Cheng et al., 2015; Tao et al., 2014b;
2014c; Xue et al., 2010; Zhang et al., 2015). They found the abundance of
hygroscopic $NH_4NO_3$ and $(NH_4)_2SO_4$ in $PM_{2.5}$ and their characteristics were the
important reason visibility reduction. However, few studies have analyzed the size
distribution of aerosol light scattering or quantified the contributions of different
emission source categories to the aerosol light scattering, particularly at the varied air
pollution levels. The roles of particles of different sizes and origins on visibility
degradation remained unclear. To fill this knowledge gap, this study conducted
campaigns at three multiple-functional sites in Nanjing, a mega city located in eastern
China. Nanjing suffered relatively heavy aerosol pollution in the Yangtze River Delta
(YRD) attrited to the massive emissions of anthropogenic air pollutants (Zhao et al.,
2015). The mixed sources of primary aerosols (e.g., coal burning) and secondary
aerosol precursors (e.g., vehicle and petrochemical industry) make Nanjing a typical
city to study the multiple influential factors of aerosol light scattering (Chen et al.,
2019). Combining online and offline techniques at different functional regions, the
IMPROVE algorithm was optimized taking the light-absorption OM into account.
The influences of aerosol size distributions and pollution levels on the aerosol
scattering effect were quantitatively evaluated based on comprehensive analysis of the
chemical compositions of particles by size and location. To explore the reasons for the



visibility reduction in different functional regions, a new coupled PMF-Mie model
was developed and the source apportionments of aerosol light scattering were
determined for the clean and polluted periods.

## 2 METHODOLOGY

### 2.1 Site description

The campaigns were conducted at three sites in Nanjing, i.e., NJU, PAES and
NUIST, representative for the suburban, urban and industrial region, respectively (see
the site locations in Figure S1 in the Supplement). NJU (32.07°N, 118.57°E) was on
the roof (25 m above the ground) of the School of the Environment building in the
Nanjing University campus in eastern suburban Nanjing (Chen et al., 2017; 2019).
PAES (32.03°N, 118.44°E) was on the roof (30 m above the ground) of the Jiangsu
Provincial Academy of Environmental Science building in western urban Nanjing.
The site was surrounded by heavy traffic and commercial and residential buildings (Li
et al., 2015). NUIST (32.21°N, 118.72°E) was on the roof of the School of the
Environment building in the Nanjing University of Information Science &
Technology campus. It was an industrial pollution site influenced by the nearby power,
iron & steel, and petrochemical industry plants (Wang et al., 2016a).

### 2.2 Aerosol sampling and chemical analysis

Pre-combusted (at 500 °C for ~5 h) quartz filters (90 mm in diameter, Whatman
International Ltd., UK) were applied for $PM_{2.5}$ sampling. The filter samples were
weighed before and after sampling under the constant temperature (23±2°C) and RH
(40±3%) for 24 hours conditioning. All the $PM_{2.5}$ samples were collected using the
TH-150C sampler (Wuhan Tianhong Ltd., China) at a flow rate of 100 L/min. From
November 2015 to March 2017, 282 daily $PM_{2.5}$ samples at the three sites (174 for
NJU, 45 for PAES and 63 for NUIST) were collected.
Three sets of ten-stage Micro-Orifice Uniform Deposit Impactors (MOUDI,
Model 110, MSP Corp., USA) were adopted to collect size-segregated particles. The
50% cutoff points of the MOUDI-110 were 18, 10, 5.6, 3.2, 1.8, 1.0, 0.56, 0.32, 0.18



and 0.056 μm. Loaded with Teflon and quartz filters (47 mm in diameter, Whatman
International Ltd., UK), MOUDI was operated at a flow rate of 30 L/min. To obtain
sufficient particles at each stage for the chemical analysis, every sampling lasted
continuously for 24 h from 9:00am. Seventy-five sets of particle samples were
obtained from December 2015 to February 2017 at NJU, 25 sets were obtained from
August 2016 to January 2017 at PAES, and 31 sets were obtained from July 2016 to
February 2017 at NUIST.
Three anions ($SO_4^{2-}$, $NO_3^-$ and $Cl^-$) and five cations ($Na^+$, $NH_4^+$, $K^+$, $Mg^{2+}$, and
$Ca^{2+}$) in particles were measured in the extracted solution of the filter samples with
ion chromatography (Dx-120, Dionex Ltd., USA). CS12A column (Dionex Ltd.) with
20 mM MSA eluent and AS11-HC column (Dionex Corp.) with 8 mM KOH eluent
were used to measured cations and anions, respectively (Chen et al., 2019). Elemental
carbon (EC) and organic carbon (OC) were measured with an OC-EC aerosol
analyzer (Sunset Inc., USA) following the thermal-optical transmittance (TOT)
protocol. More details on the analyzer operation were described in our previous
studies (Chen et al., 2017; 2019). Recent studies indicated that methanol soluble
organic carbon (MSOC) was a more suitable BrC surrogate than water soluble organic
carbon (WSOC) and was thus used in present study (Cheng et al., 2016; 2017; Huang
et al., 2018; Lei et al., 2018). The analytical procedure was described in details in
Chen et al. (2019). Elements of size-resolved particles collected in the Teflon filters
(As, Al, Ba, Cd, Co, Cr, Cu, Fe, Mn, Mo, Ni, Ti, V, and Zn) were measured with an
inductively coupled plasma-mass spectrometer (ICP-MS, PerkinElmer ELAN 9000,
USA) in order to provide further information on the aerosol sources. More detailed
information on the instrument was provided by Khan et al. (2016) including the
precision, calibration, detection limit, and analytical procedures.

**2.3 Measurements of real time aerosol scattering coefficients**
The aerosol scattering coefficients were measured using two different types of
integrating nephelometers, i.e., Aurora 1000G (Ecotech Pty Ltd., Australia) at NJU
and PAES, and Model 3563 (TSI Inc., USA) at NUIST. To obtain the dry aerosol





scattering coefficient, the three nephelometers controlled the RH of the inflow air
under 50% by the heated inlet to mitigate the impact of water vapor on the scattering
coefficient. The nephelometers at NJU and PAES were operated at a flow rate of 5
L/min, and that at NUIST was at 20 L/min. Routine maintenance including zero
calibration and span check was conducted following the instrument manual.
To explore the RH impact on aerosol light scattering, an online monitoring
instrument Cavity Attenuated Phase Shift Albedo monitor (CAPS, Shoreline Science
Research Inc., Japan) was used to measure the ambient scattering coefficient at NJU
in real ambient conditions. The instrument operates at the wavelength of 530 nm
(Onasch et al., 2015; Petzold et al., 2013), and more details on its operation during the
campaigns were provided by Chen et al. (2019).

**2.4 Data analysis**
**2.4.1 Estimation of the scattering coefficient of aerosol chemical species with**
**different methods**
The details of IMPROVE1999 and IMPROVE2007 are summarized in the
Supplement Section A1. Neglecting the light-absorbing effect of BrC, the two
algorithms could overestimate the scattering coefficient of OM (Yan et al., 2014). The
major difference between the two versions is that the IMPROVE2007 algorithm
considers the variety of mass scattering efficiencies due to particle size for $(NH_4)_2SO_4$,
$NH_4NO_3$ and OM. In this study, multiple linear regressions between the measured
light scattering components and aerosol scattering coefficients were conducted to
obtain the mass scattering efficiency (MSE) considering the presence of BrC. The
measured scattering coefficients were subtracted from the scattering coefficients of
sea salt, soil dust and coarse particles. The $PM_{2.5}$ scattering coefficient can be
estimated statistically based on the concentrations of individual chemical species as
Eq. (1):





$$
\begin{aligned}
b_{sca} = &a \times f_S(RH)[Small\ (NH_4)_2SO_4] + b \times f_L(RH)[Large\ (NH_4)_2SO_4] \\
&+ c \times f_S(RH)[Small\ NH_4NO_3] + d \times f_L(RH)[Large\ NH_4NO_3] \\
&+ e \times ([Small\ OM] - m \times [Small\ MSOC]) \\
&+ f \times ([Large\ OM] - n \times [Large\ MSOC])
\end{aligned}
\tag{1}
$$

where $b_{sca}$ is the measured $PM_{2.5}$ scattering coefficient; $a$, $c$ and $e$ are the MSEs of
$(NH_4)_2SO_4$, $NH_4NO_3$ and $OM$ (except for light-absorbing BrC) in the small size mode,
respectively; $b$, $d$ and $f$ are the MSEs of $(NH_4)_2SO_4$, $NH_4NO_3$ and $OM$ (except for
light-absorbing BrC) in the large size mode, respectively (definitions of small and
large size modes for various aerosol components can be referred to Malm et al.
(2007)); $m$ and $n$ indicate the mass fractions of light-absorbing BrC to total MSOC in
small and large modes, respectively; $f(RH)$ (including $f_L(RH)$ and $f_S(RH)$) of sulfate
and nitrate indicate the scattering hygroscopic growth factor under a given relative
humidity ($RH$), obtained from Pitchford et al. (2007).
In addition to $PM_{2.5}$, the scattering coefficient for particles at a given size
($b_{sca}(RH)$) is calculated with the Mie theory (Bohren et al., 1998; Cheng et al., 2015):
$$
b_{sca}(RH) = \int \pi [D_p \times \frac{g(RH)}{2}]^2 \times Q_{sca}[m(RH),\ Dp,\ \lambda] \times N(D_p) \times g(RH)dD_p
\tag{2}
$$

where $m(RH)$ is the aerosol refractive index; $g(RH)$ is the hygroscopic growth factor;
$Q_{sca}$ is the scattering efficiency for a single spherical particle with diameter $D_p$ and
can be calculated with the Mie theory by inputting $D_p$, $m(RH)$ and the incident
wavelength ($\lambda$); $N(D_p)$ is the number concentration of particle with diameter $D_p$. In
general, three typical models are proposed to represent the particle mixing state
including internal, external and core-shell mixture (Jacobson, 2001; Seinfeld and
Pandis, 2006). The methods of calculating the parameters including $m(RH)$ and $N(D_p)$
are different for the three mixed states, and the details can be referred to Ding et al.

(2015).

**2.4.2 Source apportionment of aerosol scattering coefficients with a coupled**
**model of PMF and Mie theory**
Positive matrix factorization (PMF) is an effective technical method for source
apportionment of atmospheric aerosols (Kim and Hopke, 2004). In this study, PMF
5.0 software was applied in the source apportionment of accumulated mode particles.





In total, 245, 145 and 163 aerosol samples were analyzed at NJU, PAES and NUIST,
respectively. It is currently difficult to resolve the sources of secondary organic
aerosol (SOA) with PMF. In this study, a simplified method was applied to
differentiate the sources of primary and secondary aerosols. Organic carbon is split
into primary and secondary organic carbon (POC and SOC), and the SOC
concentration was calculated with the EC-tracer method (Chen et al., 2017). The
source contributions of primary particles were obtained using the PMF model, and
those of secondary inorganic aerosol (SIA) and SOA were further determined based
on estimates of the nitrogen oxides ($NO_x$), sulfur dioxide ($SO_2$) and volatile organic
compounds (VOCs) emissions in a local inventory (Huang, 2018; Lang et al., 2017;
Wang et al., 2015). The chemical components applied in the PMF model included
inorganic ions, carbonaceous components and metallic elements. We followed the
method described in the PMF manual and Tian et al. (2016) to calculate the chemical
component uncertainties in the measurement dataset. Criteria including the optimum
number of factors and the minimization of an objective function Q were determined
based on the principles described in previous studies (Moon et al., 2008; Tian et al.,
2016; Watson et al., 2015) and applied in the model to obtain the best PMF solution.

A coupled model combining PMF and the Mie theory was developed to evaluate

the sources of aerosol light scattering. The procedure of the method was as follows: (1)
the EPA-PMF model was applied to quantify the contributions of different sources to
the mass concentrations of chemical species in size-segregated particles; (2) the
contribution (%) of the $i^{th}$ chemical component to the aerosol scattering coefficient at
size $D_p$ was estimated based on Mie theory; (3) the percentage contribution (%) of the
$i^{th}$ component in the $j^{th}$ source category to the total scattering at size $D_p$ was calculated
as the product of the percentage contribution (%) of the $i^{th}$ chemical species to the
total scattering and that of the $j^{th}$ source category to the mass concentration of the $i^{th}$
species in the particles at size $D_p$, as indicated in Eq.(3); and (4) the percentage
contribution (%) of the $j^{th}$ source to the total scattering at size $D_p$ was estimated by
summing $\eta_{ijD_p}$, as shown in Eq. (4).


$\qquad \eta_{ijDp} = a_{ijDp} \Box \dfrac{b_{iDp}}{\sum\limits_{i=1}^{m} b_{iDp}} \times 100\%$  (3)
$\qquad \eta_{jD_p} = \sum\limits_{i=1}^{I} \eta_{ijD_p}$  (4)
where $i$ and $j$ stand for the numbers of aerosol chemical components and potential
sources, respectively; $\eta_{ijDp}$ (%) is the contribution (%) of $i^{th}$ scattering component in
the $j^{th}$ source to the total particle scattering at size $D_p$; $\eta_j$ (%) is the contribution (%) of
the $j^{th}$ source to the total scattering at size $D_p$; $a_{ijDp}$ is the relative contribution (%) of
the $j^{th}$ source to the $i^{th}$ chemical component in particles with size $D_p$ from PMF
modeling; and $b_{iDp}$ is the contribution of the $i^{th}$ chemical component to the total
scattering from Mie modeling.

**3 RESULTS AND DISCUSSION**

**3.1 Mass concentrations and size distributions of PM compositions**

$\qquad$ Based on the national definition on ambient Air Quality Index (AQI) (MEP,
2012), we divided the whole sampling period into three categories, i.e., the clean
period with AQI less than 100, the lightly polluted period with AQI between 100 and
200, and the heavily polluted period with AQI above 200. Note that the AQI is a
unitless index calculated based on the daily concentrations of regulated air pollutants
including $NO_2$, $SO_2$, $CO$, $O_3$, $PM_{2.5}$ and $PM_{10}$ (MEP, 2012). As summarized in Table 1,
the average daily $PM_{2.5}$ mass concentrations at the three conditions were calculated at
$47.9 \pm 15.8$, $102.1 \pm 16.4$, and $163.1 \pm 13.6$ μg/m$^3$, respectively. The mass
concentration of secondary inorganic ions ($SO_4^{2-}$, $NO_3^-$ and $NH_4^+$) for the heavily
polluted period was 4.4 and 2.2 times those for the clean and lightly polluted periods,
respectively. The corresponding values for the carbonaceous aerosols (the sum of OC
and EC) were 3.1 and 1.9 times, respectively, and the OC to EC ratios increased from
4.5 for the clean period to 5.2 for the heavy period. In addition to the particulate
components, gaseous pollutants such as $NO_2$ and $SO_2$ were also significantly elevated
from the clean to the heavy periods. These results imply that secondary aerosol





formation was an important source of enhanced $PM_{2.5}$ for the heavily polluted period.

Figure S2 in the Supplement compares the size distributions of mass

concentrations for particles and selected chemical components under three pollution
levels. Bimodal size distributions were found for PM and OC mass concentrations,
with the two peaks at the ranges of 0.56-1.0 μm and 3.2-5.6 μm, respectively. This
bimodal pattern could partly result from the coexistence of primary and secondary
sources of OC. POC with larger sizes may contribute largely to the peak in the coarser
particles. In contrast, due to chemistry reactions of biogenic and anthropogenic VOCs,
SOC was expected to be abundant in the accumulation mode (0.18-1.8 μm) (Cao et al.,
2007). The size distributions of $NO_3^-$ and $SO_4^{2-}$ followed a unimodal distribution with
the mass concentrations peak at the range of 0.56-1.0 μm, as most of the inorganic
aerosols were generated through secondary formation. The mass concentrations of
PM, $NO_3^-$, $SO_4^{2-}$ and OC for all sizes were enhanced from the clean to the polluted
periods, and the biggest differences were found in the size bin of 0.56-1.0 μm. As
shown in Figure S2a, the concentrations of $PM_{0.56-1.0}$ for the heavily and lightly
pollution periods were 7.0 and 2.7 times greater than that for the clean period,
respectively. Moreover, $PM_{0.56-1.0}$ contributed 31%, 23%, and 15% to the total mass
concentrations of particles for the heavily, lightly polluted and clean periods,
respectively, implying that the enhanced concentration of $PM_{0.56-1.0}$ was an important
reason for the aggravated pollution. As shown in Figure S2b-S2d, the sum of $NO_3^-$,
$SO_4^{2-}$ and OC for the heavily polluted period was 10.7 and 2.9 times greater than those
for the lightly polluted and clean periods, respectively. From clean to heavily polluted
periods, the collective mass fraction of the three components to $PM_{0.56-1.0}$ increased
from 42% to 64%. The results indicated that the increased $NO_3^-$, $SO_4^{2-}$ and OC at the
size bin of 0.56-1.0 μm could be an indicator for the serious air pollution events.

To explore the mass fractions of major chemical species in the particles, the PM

mass was reconstructed as $(NH_4)_2SO_4$ (1.38×$SO_4^{2-}$), $NH_4NO_3$ (1.29×$NO_3^-$), OM
(1.55×OC), fine soil (FS) and EC (Cheng et al., 2015; Pitchford et al., 2007). As
shown in Figure S3 in the Supplement, strong correlations were found between the
reconstructed PM mass concentrations and the measurements for $PM_{1.8}$ ($R^2$=0.85) and



$PM_{10}$ ($R^2$=0.81) at the three sites. The slope of $PM_{1.8}$ (0.80) was greater than that of
$PM_{10}$ (0.65), indicating smaller unidentified fraction in the $PM_{1.8}$. The larger
unidentified mass in the reconstructed $PM_{10}$ was probably due to underestimation in
the crustal components (Hueglin et al., 2005).
Figure 1 presents the mass concentrations and fractions of the reconstructed
aerosol chemical species by particle size under the three pollution levels. $NH_4NO_3$,
$(NH_4)_2SO_4$, and OM were the dominant components in particles. From the clean to
heavily polluted periods, their mass fractions to $PM_{1.8}$ increased from 16.9 to 35.3%,
from 14.9 to 28.6% and from 16.7 to 22.2%, respectively (Figure 1b, Figure 1d and
Figure 1f). The mass fraction of OM in $PM_{1.8}$ was 5.4% and 7.4% larger than
$NH_4NO_3$ and $(NH_4)_2SO_4$ for the clean period, while 13.3% and 6.6% smaller than
those for the heavily polluted period, respectively. The results further confirmed that
substantial growth in the mass of $NH_4NO_3$ and $(NH_4)_2SO_4$ was an important reason
for the aerosol pollution. The formation of sulfate, nitrate, and ammonium (SNA) is
mainly affected by the emissions of precursors and the atmospheric oxidation capacity
Due to the great use of fossil fuel consumption, the emissions of precursors $SO_2$ and
$NO_X$ per unit area in eastern China were estimated 2.3 and 3.4 times larger than the
national average, respectively (Cheng et al., 2012; Shi et al., 2014). Under high RH,
moreover, the SNA formation could significantly be elevated through gas-to-particle
heterogeneous reactions for the heavily polluted period (Seinfeld and Pandis, 2006).
Sulfate mass concentration, for example, increased from 6.4 μg/m$^3$ for the clean
period to 53.3 μg/m$^3$ for the heavily polluted period. Among all the size bins,
$NH_4NO_3$ and $(NH_4)_2SO_4$ were estimated to contribute the most to the mass
concentrations for 0.56-1.0 μm particles, with their mass fraction ranging 19.4-39.7%
and 18.1-34.7%, respectively, across different pollution levels. In comparison, the
largest contributions of OM appeared in the 0.056-0.18 μm fraction and were 31.2%,
29.0% and 52.3% for the clean, lightly polluted and heavily polluted periods,
respectively. As the largest PM fraction was found in the 0.56-1.0 μm size bin for the
heavily pollution period, the elevated concentrations of $NH_4NO_3$ and $(NH_4)_2SO_4$ in
$PM_{0.56-1.0}$ were the major causes of the increased aerosol pollution.



Figure S4 in the Supplement compares the size distributions of PM mass
concentrations and selected chemical species at the three sites. As mentioned above, a
bimodal distribution with two peaks at 0.56-1.0 μm and 3.2-5.6 μm was observed for
PM and OC at all the three sites, attributed to the coexistence of primary and
secondary sources. Different from PAES and NUIST, $NO_3^-$ had an obvious small
coarse mode peak at NJU. Previous studies suggested that the chemistry of coarse
mode $NO_3^-$ can vary in different locations, and the components include $NH_4NO_3$,
$NaNO_3$ and $Ca(NO_3)_2$ (Pakkanen et al., 1996). As NJU was close to the G25 highway,
the reaction of $HNO_3$ with crustal particles could be an important process for coarse
mode $NO_3^-$ formation. The highest mean concentrations of $NO_3^-$ and $SO_4^{2-}$ at the
0.56-1.0 μm size among the three sites were observed at NJU, followed by NUIST
and PAES. As $NO_3^-$ and $SO_4^{2-}$ were the major components of the aerosol light
scattering, the variety of their mass concentrations at 0.56-1.0 μm could be a crucial
reason for the visibility difference among the three sites. A greater difference was
found for the size distribution of OC among the three sites, and the highest
concentration at the 0.56-1.8 μm size was observed at NUIST. Our previous work
found that NUIST was greatly influenced by VOCs emissions of surrounding
industrial plants (Chen et al., 2019). Given its capability of light scattering and
absorption, the abundant OC in the area could play an important role on the visibility.

**3.2 Evaluation and optimization of the IMPROVE algorithm**

Figure S5 in the Supplement presents the linear regressions between the
measured daily aerosol scattering coefficients with the CAPS ($b_{sp-m}$) and those
calculated with IMPROVE algorithms ($b_{sp-1999}$ and $b_{sp-2007}$) based on the measured
concentrations of particle components at the three sites. At each site, strong
correlations were found between the observation and IMPROVE estimation ($R^2 \geq$
0.94), indicating consistency between the different techniques. As shown in Figure
S5a, the calculated aerosol scattering coefficients $b_{sp-1999}$ were 30%, 16% and 19%
smaller than the measured values at NJU, PAES and NUIST, respectively. Similar
results were found for other megacities in eastern China. Based on the online





analytical methods, for example, Cheng et al. (2015) estimated that the scattering
coefficients predicted by the IMPROVE1999 algorithm were 34% smaller than the
measurement for a heavy pollution period in Shanghai. A greater underestimation of
the scattering coefficient existed at NJU than the other two sites, partly due to the
relatively abundance of sulfate and nitrate in particles at NJU. The sum of $SO_4^{2-}$ and
$NO_3^-$ accounted for 35.3 ± 13.2% of the total mass concentrations of $PM_{2.5}$ at NJU,
larger than the fraction at PAES (27.6 ± 12.9%) and NUIST (24.1 ± 11.6%) (note the
$SO_4^{2-}$ and $NO_3^-$ concentrations at the 0.56-1.0 μm were the largest at NJU as well, as
shown in Figure S4). Tao et al. (2014b) and Cheng et al. (2015) suggested that the
relatively small MSE of sulfate and nitrate aerosols in the IMPROVE1999 algorithm
might result in underestimation of the scattering coefficient in China, as sulfate and
nitrate were the main light-scattering components in $PM_{2.5}$.
As shown in Figure S5b, $b_{sp-2007}$ was only 4% smaller than the measurement at
NJU, and 4% and 18% larger at PAES and NUIST, respectively. Overall, the
performance of the IMPROVE2007 algorithm was better than that of the
IMPROVE1999, although deviation still existed due to the uncertainty in MSEs for
chemical species and the presence of light absorption organic matter such as BrC. A
relatively large deviation between $b_{sp-m}$ and $b_{sp-2007}$ was found at NUIST compared to
NJU and PAES. Chen et al. (2019) and Shao et al. (2016) found higher annual average
concentration of non-methane hydrocarbon at NUIST (34.4 ppbv) than NJU (22.0
ppbv) or PAES (27.1 ppbv). The more VOCs in the atmosphere were expected to
increase the SOC formation and to result in big deviation of $b_{sp-2007}$, as the OM with
light-absorption capability was not considered in IMPROVE2007.
Using the optimized IMPROVE algorithm as described in Section 2.4.1, the
aerosol scattering coefficients were recalculated and compared against the observation
at the three sites, as illustrated in Figure 2. Good correlations were found between the
observed and calculated scattering coefficients at all the sites ($R^2 \geq 0.96$), and the
regression slopes were estimated to be much closer to 1 than those between
observations and predictions with the IMPROVE1999 or IMPROVE2007 algorithms
(Figure S5). In addition, the MSEs calculated based on the Mie theory were applied to



evaluate the results of the IMPROVE algorithms. As presented in Figure S6 in the
Supplement, the MSEs of $(NH_4)_2SO_4$ and $NH_4NO_3$ calculated with the optimized
IMPROVE algorithm were closer to the MSE simulated by Mie theory than those
with the IMPROVE2007 algorithm. The results indicated the optimized algorithm had
a better performance and could reduce the bias from the US IMPROVE algorithm.
As summarized in Table 2, the MSEs estimated with the optimized IMPROVE
algorithm were 2.29, 4.82, 2.62, 5.35, 4.46, and 6.41 $m^2/g$ for small sulfate, large
sulfate, small nitrate, large nitrate, small and large OM, respectively. In comparison,
the MSEs for the small and large size modes using the IMPROVE2007 were 2.2 and
4.8 $m^2/g$ for $(NH_4)_2SO_4$, respectively, and 2.4 and 5.1 $m^2/g$ for $NH_4NO_3$, respectively.
The slightly larger MSEs from the optimized IMPROVE algorithm for $(NH_4)_2SO_4$ and
$NH_4NO_3$ implied underestimation of the scattering coefficients of inorganic
components when applying the previous algorithm. There were clear differences in
the MSEs of OM (especially for fine OM) between the two algorithms, resulting from
consideration of the light-absorbed OM in the optimized algorithm. Indicated by the
m values in Table 2, the light-absorbed OC accounted for 66% and 71% of the fine
MSOC mass at NJU and PAES, respectively, indicating that most of the fine MSOC
had only light-absorption capacity. Unlike NJU and PAES, less than half of the fine
MSOC (39%) had light-absorption capacity at NUIST, likely resulting from the varied
sources of OM at the three sites. As described in our previous study (Chen et al.,
2019), substantial OC was from the secondary formation in industrial polluted region,
and its light-absorption capacity was weaker than that from the primary emissions.
Through field measurement and data reconstruction in different cities, previous
studies explored the concentrations of $PM_{2.5}$ and its chemical components for various
cities in China (Chen et al., 2019; Feng et al., 2012; Lai et al., 2016; Tao et al., 2013.,
Yang et al., 2011; Zhao et al., 2013). The major components of light scattering in
aerosols, SNA, was found to typically account for half of the $PM_{2.5}$ mass
concentrations in eastern Chinese cities like Nanjing, Shanghai, and Jinan (Yang et al.,
2011). Given the similar level and strong regional transport of pollution among those
cities, the optimized IMPROVE algorithm applied in Nanjing in this work is believed





to be more suitable than the previous algorithms for eastern China.

### 3.3 Effects of mixing state and relative humidity on aerosol light scattering

Figure 3 presents the scattering coefficients measured by nephelometer and those
simulated by Mie theory at the three sites under dry conditions (RH < 40%). The
simulated scattering coefficients based on the assumption of an external mixing state
were larger than those based on core-shell and internal mixing states at all the three
sites. Compared with the internal and core-shell states, the simulated scattering
coefficients in the external mixing state were closer to the measurements at NJU and
PAES (Figure 3a and 3b), indicating the reasonable assumption of external mixtures
as the main mixing state of particles. Similarly, Ma et al. (2012) also suggested that
the external mixture was an important particle mixing state in northern China based
on a stochastic particle-resolved aerosol box model. Assuming the aerosol
components were externally mixed, Cheng et al. (2015) estimated the MSEs of
aerosol species in Shanghai, and found better agreement between the optimized
scattering coefficients and the measurements. At NUIST, the measured scattering
coefficients were closer to the simulated values in internal and core-shell states, likely
due to the high aging level of SOA at the industrial site (Figure 3c). Due to the strong
atmospheric oxidation and thereby the abundance of SOA coatings at NUIST, our
previous study suggested that the aerosol aging process could result in the growth of
internally mixed BC (Chen et al., 2019). Based on the observation of $O_3$ and
percentage of internally mixed BC, Lan et al. (2013) suggested that photochemical
production of secondary aerosol components was the main reason for the switching
from an external mixing state to an internal mixing state for BC.
In an actual environment, ambient aerosols are typically hygroscopic under the
conditions of high RH, and it is an important reason for visibility degradation. Table
S1 in the Supplement summarizes the growth factors (GF) of particle size measured in
Nanjing at different RH levels in previous studies. To evaluate the rationality of those
GF values, we followed the method by Tao et al. (2014b) and calculated the scattering
hygroscopic growth factor (f(RH)) at NJU based on the measured ambient scattering





coefficients by CAPS and the dry scattering coefficients by nephelometer, as shown in
Figure S7 in the Supplement. The correlation between f(RH) and RH was fitted
through the power regression. Figure S8 in the Supplement presents a good agreement
between the scattering coefficients estimated by f(RH) and those obtained by the Mie
theory ($R^2$=0.95). The results indicate the accuracy of the GF values applied on
different particle sizes and RH levels. The estimated and measured scattering
coefficients at NJU under ambient condition are shown in Figure S9 in the
Supplement. Different from the estimation under the dry conditions, the lowest value
was found for the externally mixing state among the three mixing modes. In the
externally mixing state, only sulfate and nitrate particles had hygroscopicity under wet
conditions, whereas each particle had the capability of hygroscopic growth in the
internal mixing and core-shell states, resulting in a significant increment in the
scattering coefficient. Similarly, comparing the measured scattering coefficients under
the dry and ambient conditions (Figure 3 and Figure S9), the simulated values based
on an external mixing state were closer to the measurements than the other two modes,
implying that RH had a limited effect on the particle mixing state.

To explore the impact of RH on the light scattering of particles with different

sizes, the size distribution of f(RH) was estimate and shown in Figure 4. Large
differences were found between f(RH) when the RH was above and below 75%, and
high RH enhanced the capacity of scattering hygroscopicity growth of small size
particles. Approximately 140 nm particles had strong hygroscopicity when the RH
was below 75%, whereas a high f(RH) (1.41±0.03) was observed for the
accumulation mode particles from 100 to 400 nm when the RH was above 75%.
Similar results were reported for Beijing: larger hygroscopic GF was measured for
accumulation mode particles (100–300 nm) with a hygroscopicity tandem differential
mobility analyzer (H-TDMA), consistent with the elevated abundance of the
light-scattering compositions such as sulfate and nitrate (Meier et al., 2009).

**3.4 Size distribution of aerosol light scattering by pollution level**

Figure 5 shows the size distribution of the scattering coefficients for particles and



given chemical components under the three pollution levels. The scattering
coefficients of particles for all size categories were the largest for the heavily polluted
period (Figure 5a). The accumulation mode particles (0.18-1.8 μm) accounted for
92.9%, 92.6 and 93.4% of the total scattering coefficients for the clean, lightly
polluted and heavily polluted periods, respectively. In particular, particles in the size
bin of 0.56-1.0 μm accounted for 57% and 63% of the scattering coefficient for the
heavily and lightly polluted periods, respectively, much larger than that for the clean
period, 38%. From the results of Section 3.1, the abundance of particles of different
sizes was considered to be an important factor for the variety of scattering coefficients
across the whole size range.
As the dominant chemical components of aerosol light scattering, $(NH_4)_2SO_4$,
$NH_4NO_3$ and OM collectively contributed 90%, 76% and 60% to the mass
concentrations of $PM_{0.56-1.0}$ for the heavily polluted, lightly polluted and clean periods,
respectively (Figure S2b-S2d). The scattering coefficients of $(NH_4)_2SO_4$ and $NH_4NO_3$
were the largest in the size bin of 0.56-1.0 μm for the three pollution levels, and their
contributions increased along with the aggravation of pollution (Figure 5b and Figure
5c). The OM concentration in the size bin of 0.56-1.0 μm was 2.5 $μg/m^3$ for the clean
period, and those for the lightly and heavily polluted periods were 160% and 510%
larger, respectively. The scattering coefficient of OM in the size bin of 0.56-1.0 μm
for the heavily polluted period was 15% less than that for the lightly polluted period,
indicating the more important role of OM in the particle scattering effect for the
lightly polluted period (Figure 5d). The large OM scattering contribution could likely
be explained by the elevated mass fraction of OM and/or enhancement of the OM
MSE. It could be inferred that the low visibility during heavy pollution resulted
mainly from the enhancement of the scattering effect of SNA.
The MSEs of given chemical components in $PM_{1.8}$ are presented by pollution
level in Figure 6. Increased MSEs for $(NH_4)_2SO_4$ and $NH_4NO_3$ were found along with
the elevated $PM_{2.5}$ pollutions (Figure 6a). The large contributions of inorganic
components and their strong light scattering ability were important reason for the
reduced visibility during the heavily polluted period. Although the largest OM



concentrations were observed in each size bin for the heavy pollution period, the smallest MSE of OM in $PM_{1.8}$ was found for the heavily polluted period (3.73 $m^2/g$, Figure 6b). As discussed in Section 3.2, most of the fine MSOC was expected to have only a light-absorption effect whereas large MSOC had light-scattering capability. With the optimized IMPROVE algorithm, the mass fraction of light-absorption OC to total MSOC mass was estimated at 66.9±5.8% for the heavily polluted period, much larger than those for clean and lightly polluted periods at 44.3±6.5% and 50.8±5.9%, respectively, as shown in Figure S10 in the Supplement. Therefore, the small MSE of OM for the heavily polluted period was partly attributed to the abundance of light absorption BrC in $PM_{2.5}$.

For the whole research period, the MSEs of $(NH_4)_2SO_4$, $NH_4NO_3$ and OM in $PM_{1.8}$ were calculated at 3.95, 4.26 and 4.14 $m^2/g$ with the Mien theory, while the analogue numbers in $PM_{2.5}$ were 3.94, 4.31 and 5.25 with the optimized IMPROVE algorithm, respectively (Table 2). Very good agreement between the two methods was found for SNA, and clearer discrepancy existed for OM, indicating a larger uncertainty in the evaluation of organic aerosol scattering.

**3.5 Source apportionment of aerosol light scattering with the PMF-Mie coupled model**

As illustrated in Figure 5, the light scattering of the accumulation mode (0.18-1.8 μm) accounted for the largest proportion of the total light scattering. To better understand the causes of visibility degradation, the source apportionment of aerosol light scattering at this size range was conducted for different pollution levels with the PMF and Mie coupled model, as described in Section 2.4.2. The PMF model was adopted to identify the potential sources and to estimate their respective contributions to the mass concentration of accumulation mode particles. To resolve the appropriate number of factors, different numbers of identifiable sources were tested. The results of the source profiles and their contributions to accumulation mode particles at the three sites are presented in Figure S11 in the Supplement and Figure 7a-c, respectively. The main sources identified at the three sites include coal combustion, industrial



pollution, vehicle, fugitive dust, biomass burning, and SIA (Figure S11). Compared to
NJU and NUIST, vehicle contributed more to accumulated particles at the urban site
PAES (Figure 7b). As stated in Section 2.4, we assumed that the contribution of the
individual source category to the secondary particle component was proportional to
the fraction of that source category to the emissions of corresponding precursors
(Lang et al., 2017). Based on the emission inventory of precursors of SOC (VOCs)
and SIA ($NO_x$, $SO_2$ and $NH_3$) in Nanjing (Huang, 2018), the source apportionment for
primary and secondary particles of accumulation mode at the three sites were
estimated, and the results are presented in Tables S2-S4 in the Supplement. With the
source apportionment of secondary components, the contributions of coal combustion
and industrial pollution increased 45-50% and 138-478% compared to those for
primary particles across the three sites, respectively. The result indicates that the
gaseous precursors from coal combustion and industrial pollution greatly elevate the
aerosol pollution.

The contributions of different aerosol species to the aerosol light scattering were

estimated using the Mie model, and the results are presented in Table S5 in the
Supplement. OM contributed the most to the total scattering at the three sites (31%,
29% and 33% for NJU, PAES and NUIST, respectively). Compared to other Chinese
mega cities, the contribution of OM in Nanjing was close to that for inland cities like
Beijing (Tao et al., 2015) and Tianjin (Wang et al., 2016b), but was much larger than
that observed in a coastal megacity, Guangzhou (Tao et al., 2014c).

Combined with the source apportionment from the PME model, Figure 7d-7f

illustrates the source contribution to aerosol light scattering at the three sites. Coal
combustion, industrial plants and vehicle were the major sources of the aerosol light
scattering in Nanjing, and the three source categories collectively accounted for
64-70% of the total scattering capacity of aerosols. Given their relatively intensive
activities in urban and industrial regions, vehicles and industrial plants were identified
as the largest contribution sources at PAES and NUIST, respectively. Indicated by the
dashed lines in Figure 7d-7f, the collective contributions of secondary aerosol
components were estimated to be 26.7%-35.2% of the total scattering at the three sites,





suggesting the important role of secondary aerosol formation in visibility reduction.
Figure 8 illustrates the source apportionment of aerosol light scattering for the
clean and polluted periods at the three sites. Coal combustion contributed the most to
the total scattering for the clean period, and the contribution declined significantly for
the polluted period, from 39% to 21%, from 38% to 19% and from 35% to 18% at
NJU, PAES and NUIST, respectively. The results implied that coal combustion might
not be the most important reason for visibility degradation in polluted periods.
Similarly, the contribution of fugitive dust during the polluted period was estimated to
be smaller than that for the clean condition. In contrast, the contributions of vehicles
and industrial pollution to light scattering increased from 27% to 48%, from 27% to
47% and from 31% to 62% for the polluted periods compared to the clean period at
NJU, PAES, and NUIST, respectively. As shown in Figure 8b and 8c, particularly, the
contribution of primary emissions from vehicles to aerosol scattering was estimated to
increase from 11.4% to 21.5% at PAES, and that from industrial plants increased from
4.5% to 13.5% at NUIST. The primary aerosol emissions from vehicles and industrial
plants were thus identified as the main cause of visibility reduction in the urban and
industrial areas, respectively. Similarly, Wang et al. (2016b) suggested that vehicle
was the dominate source of aerosol light extinction in Hangzhou, with the
contribution to the total extinction coefficient of $PM_{2.5}$ reaching 30.2%. The present
study indicated that more effective measures for reducing the primary particle
emissions from vehicles and industrial production should be conducted to avoid
severe haze pollution in urban and industrial regions.
In addition, the results suggest that secondary aerosols were another important
contributor to the reduced visibility. From the clean to the heavily polluted periods, as
shown in Figure 8, the contributions of secondary aerosols to the total light scattering
increased from 19.9% to 36.7%, from 20.9% to 32.4%, and from 28.6% to 41.7% at
NJU, PAES, and NUIST, respectively. As shown in Table 3, the contributions of SIA
to the total scattering at the three sites were ranged at 14.5%-19.9% and 24.5%-28.0%,
much more than those of SOA at 4.1%-8.7% and 7.9%-13.7% for the clean and
polluted periods, respectively. The results imply that SIA had a greater impact on



visibility degradation. Although the contribution of coal combustion to the total scattering declined from clean to polluted periods, the contributions of SIA from coal combustion for the polluted periods were 88%, 35% and 36% larger than those for the clean period at NJU, PAES and NUIST, respectively. The enhancement of SIA from coal combustion was thus an important cause of polluted days. Moreover, the contribution of SOA to the total scattering coefficient during the polluted periods was estimated at 13.7% at NUIST, larger than the 7.9% and 9.1% at PAES and NJU, respectively, indicating that the contribution of SOA to visibility reduction at industrial polluted areas should not be ignored. Notably, there is uncertainty in the methodology of source apportionment of aerosol scattering coefficients. In particular, the assumption that the secondary components were proportional to the emissions of their precursors is a simplified method and probably led to large bias, as the complicated nonlinear mechanism of secondary aerosol formation is not recognized. More tools including chemistry transport modeling and radiocarbon measurements are thus recommended to be integrated into future studies to better determine the primary and secondary sources of aerosols.

## 4 CONCLUSIONS

A comprehensive investigation of the light-scattering properties of atmospheric aerosols was conducted from November 2015 to March 2017 at three functional sites in Nanjing. High concentrations of sulfate and nitrate in $PM_{0.56-1.0}$ were the major causes of the heavy particle pollution events. The varied abundance of secondary inorganic components at the three sites was an important reason for the visibility differences, and OC played an important role on the visibility reduction in the industrial area due to its complicated optical effect. Based on the measured aerosol scattering coefficients and the mass concentrations of aerosol components, an optimized algorithm of IMPROVE that considered the light absorption effect of OM was developed to better represent the aerosol optical property.

Compared with internal and core-shell mixing states, the simulated scattering coefficients based on an external mixing assumption were closer to the measurements





at NJU and PAES, indicating that externally mixed particles widely existed at urban and suburban areas. At the industrial site NUIST, the high aging level of SOA was the main reason for particle switching from external to internal mixing states. The results for the scattering coefficients under dry and ambient conditions indicated that RH had little effect on particle mixing state but a large impact on the scattering coefficients. Particles in the size range of 0.56-1.0 μm contributed the most (38%-63%) to the total scattering coefficient under different pollution levels. As the dominant light scattering species in aerosols, $NH_4NO_3$, $(NH_4)_2SO_4$ and OM collectively contributed 90%, 81% and 76% of the mass concentrations of $PM_{0.56-1.0}$ for the heavily polluted, lightly polluted and clean periods, respectively. The low visibility during the heavy pollution period mainly resulted from the enhanced light scattering of SNA. The abundance of light-absorption OC was an important reason for the relatively low contribution of OM to light scattering in the heavy pollution period.

Through a coupled model of PMF and Mie theory, we found coal combustion, industrial plants and vehicle were the main sources of the visibility reduction in Nanjing. Vehicles and industrial plants were the main causes for visibility reduction in urban and industrial areas, respectively. The increased emissions of SIA precursors from coal combustion were an important cause of polluted days, and the contribution of SOA to visibility reduction at industrial pollution areas should not be ignored. The source apportionment of aerosol light scattering in this work provides scientific evidence for the control of haze pollution in different functional areas of cities in developed eastern China.

**DATA AVAILABILITY**

All data in this study are available from the authors upon request.



**AUTHOR CONTRIBUTIONS**


DC developed the strategy and methodology of the work and wrote the draft. YZ
improved the methodology and revised the manuscript. JZ, HY and XY provided
observation data of aerosol scattering coefficient.

**COMPETING INTERESTS**


The authors declare that they have no conflict of interest.

**ACKNOWLEDGEMENT**


This work was sponsored by the Natural Science Foundation of China
(41922052 and 91644220), and the National Key Research and Development Program
of China (2017YFC0210106).

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





**FIGURE CAPTIONS**

**Figure 1. The mass concentrations and fractions of the main chemical components of particles with different sizes in Nanjing on clean, lightly-polluted and heavily-polluted days during the sampling period.**

**Figure 2. Linear regressions between the measured light scattering coefficients and those estimated with the optimized IMPROVE algorithm at NJU, PAES, NUIST and all three sites.**

**Figure 3. The comparison of measured and estimated dry scattering coefficients based on the assumptions of external, internal, and core-shell mixture at NJU (a), PAES (b) and NUIST (c).**

**Figure 4. The size distribution of hygroscopic scattering growth of particles under varied relative humidity levels at the three sites.**

**Figure 5. The size distribution of scattering coefficients of aerosol particles (a), $(NH_4)_2SO_4$ (b), $NH_4NO_3$ (c) and OM (d) under different pollution levels. The contributions of particles with different sizes to total scattering coefficient are indicated in the panels as well.**

**Figure 6. The size distribution of mass concentrations of $(NH_4)_2SO_4$, $NH_4NO_3$ (a), and OM (b) under different pollution levels and mass scattering efficiencies (MSE) for $PM_{1.8}$. The size of dot represents the MSEs of $PM_{1.8}$ (Unit: $m^2/g$).**

**Figure 7. Source apportionment of accumulation mode particles at NJU (a), PAES (b) and NUIST (c), and source apportionment of light scattering for accumulation mode particles at NJU (d), PAES (e) and NUIST (f). The shadow represents the contribution of secondary aerosols from each source category.**

**Figure 8. Source apportionment of light scattering for accumulation mode particles for the clean and polluted periods at NJU (a), PAES (b) and NUIST (c). The shadow represents the contribution of secondary aerosols from each source category.**



**TABLES**
**Table 1. The concentrations of particulate matter and its chemical components**
**($\mu g/m^3$), light scattering coefficients ($Mm^{-1}$), and selected meteorological**
**parameters including wind speed (WS, m/s) and relative humidity (RH, %) at all**
**the three sites for different pollution levels from November 2015 to January**
**2017.**

| Category | Clean period | Lightly polluted period | Heavily polluted period |
|---|---|---|---|
| AQI | $65.8 \pm 15.7$ | $110.6 \pm 21.3$ | $209.4 \pm 30.1$ |
| $PM_{10}$ | $80.4 \pm 26.3$ | $143.1 \pm 28.6$ | $244.2 \pm 21.2$ |
| $PM_{2.5}$ | $47.9 \pm 15.8$ | $102.1 \pm 16.4$ | $163.1 \pm 13.6$ |
| OC | $8.6 \pm 3.2$ | $14.2 \pm 3.2$ | $27.6 \pm 5.0$ |
| EC | $1.9 \pm 0.9$ | $3.0 \pm 1.2$ | $5.3 \pm 0.1$ |
| $SO_4^{2-}$ | $6.9 \pm 3.9$ | $13.5 \pm 5.6$ | $33.8 \pm 9.2$ |
| $NO_3^-$ | $10.5 \pm 5.4$ | $22.7 \pm 8.7$ | $47.9 \pm 17.7$ |
| $Cl^-$ | $1.8 \pm 1.5$ | $2.2 \pm 1.3$ | $4.8 \pm 1.4$ |
| $Ca^{2+}$ | $1.2 \pm 0.8$ | $1.3 \pm 1.6$ | $0.8 \pm 0.1$ |
| $Na^+$ | $0.8 \pm 0.2$ | $0.9 \pm 0.3$ | $1.0 \pm 0.1$ |
| $Mg^{2+}$ | $0.1 \pm 0.1$ | $0.2 \pm 0.1$ | $0.1 \pm 0.0$ |
| $NH_4^+$ | $5.1 \pm 1.9$ | $9.2 \pm 2.2$ | $16.9 \pm 2.5$ |
| $K^+$ | $0.9 \pm 0.2$ | $1.3 \pm 0.3$ | $2.1 \pm 0.7$ |
| CO | $0.8 \pm 0.2$ | $1.3 \pm 0.3$ | $1.6 \pm 0.1$ |
| $NO_2$ | $57.4 \pm 18.0$ | $71.6 \pm 20.0$ | $91.2 \pm 32.8$ |
| $SO_2$ | $17.7 \pm 6.5$ | $21.1 \pm 6.0$ | $29.5 \pm 12.5$ |
| WS | $1.6 \pm 0.3$ | $1.4 \pm 0.5$ | $1.0 \pm 0.3$ |
| RH | $56.1 \pm 13.5$ | $62.7 \pm 10.8$ | $68.9 \pm 4.9$ |
| $b_{sp}$ | $251.4 \pm 170.8$ | $558.3 \pm 236.4$ | $1286.2 \pm 293.3$ |




**Table 2. The mass scattering efficiencies (MSEs, m$^2$/g) of chemical species in the**
**optimized and the existing algorithms from the Interagency Monitoring of**
**Protected Visual Environments (IMPROVE). The sample numbers and the mass**
**fractions of light-absorption BrC to MSOC for small and large size modes (i.e., *m***
**and *n* in Eq.1) are provided for the optimized algorithm.**

| | Modes | NJU | PAES | NUIST | All the three sites | IMPROVE 2007 | IMPROVE 1999 |
|---|---|---|---|---|---|---|---|
| | small | 2.32 | 2.02 | 2.43 | 2.29 | 2.2 | - |
| MSE of $(NH_4)_2SO_4$ | large | 4.71 | 4.92 | 4.86 | 4.82 | 4.8 | - |
| | overall | 3.91 | 3.88 | 4.03 | 3.94 | - | 3 |
| | small | 2.67 | 2.48 | 2.56 | 2.62 | 2.4 | - |
| MSE for $NH_4NO_3$ | large | 5.37 | 5.31 | 5.26 | 5.35 | 5.1 | - |
| | overall | 4.41 | 4.13 | 4.23 | 4.31 | - | 3 |
| | small | 4.4 | 4.56 | 4.22 | 4.46 | 2.8 | - |
| MSE of OM | large | 6.23 | 6.36 | 6.45 | 6.41 | 6.1 | - |
| | overall | 5.26 | 5.03 | 5.35 | 5.25 | - | 4 |
| m | - | 0.66 | 0.71 | 0.39 | 0.67 | - | - |
| n | - | 0.29 | 0.27 | 0.33 | 0.31 | - | - |
| Sample number | - | 174 | 45 | 63 | 282 | - | - |




**Table 3. The source contributions of secondary aerosols to aerosol light**
**scattering at the three sites for the clean and polluted periods (%).**

| Air quality level | Sources | NJU | | PAES | | NUIST | |
|---|---|---|---|---|---|---|---|
| | | SIA | SOA | SIA | SOA | SIA | SOA |
| Clean | Coal combustion | 6.6 | 0.8 | 6.5 | 1.1 | 7.5 | 1.3 |
| | Industrial plants | 5.8 | 3.6 | 4.2 | 1.5 | 8.2 | 6.3 |
| | Vehicles | 2.1 | 1.0 | 6.1 | 1.5 | 4.2 | 1.1 |
| | Total | 19.9 | | 20.9 | | 28.6 | |
| Polluted | Coal combustion | 12.4 | 1.6 | 8.8 | 2.3 | 10.2 | 2.2 |
| | Industrial plants | 10.2 | 5.8 | 7.8 | 3 | 12.6 | 9.9 |
| | Vehicles | 5.0 | 1.7 | 7.9 | 2.6 | 5.2 | 1.6 |
| | Total | 36.7 | | 32.4 | | 41.7 | |






**Figure 1**

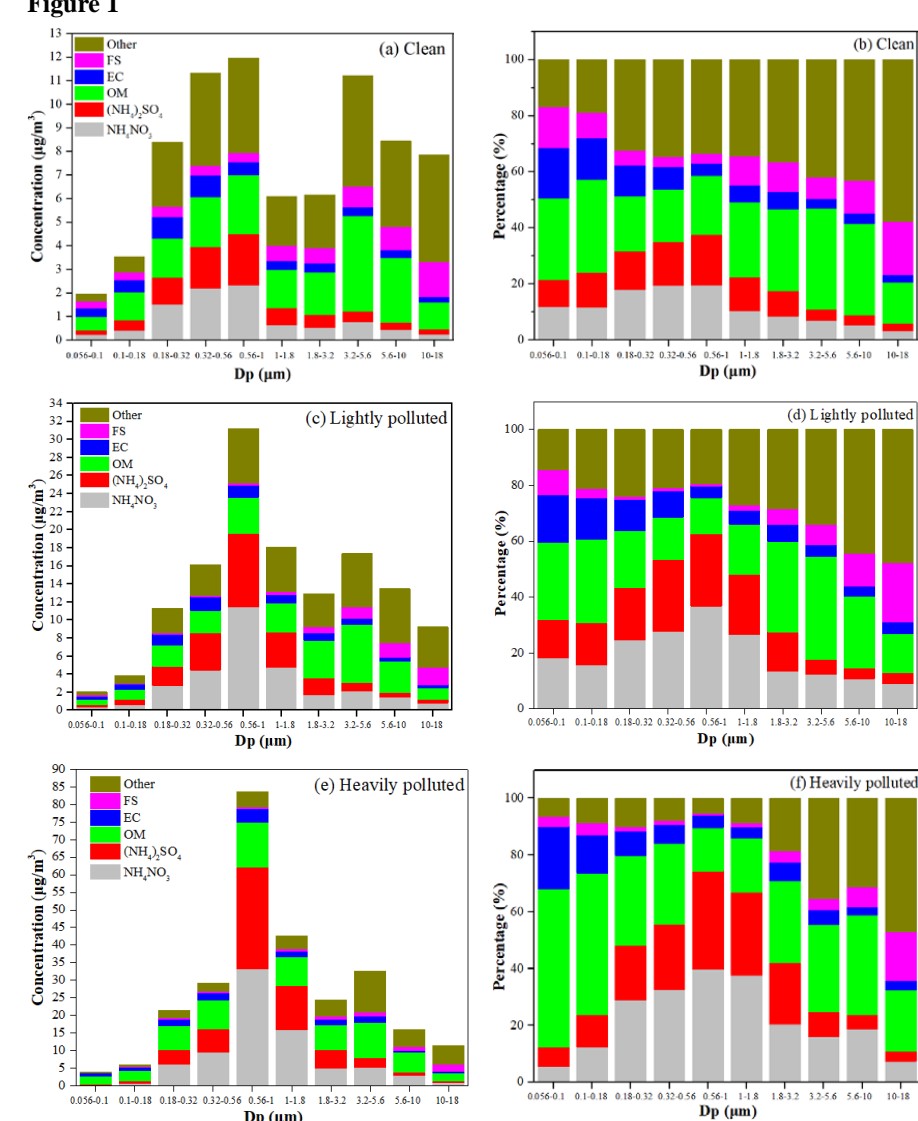



**Figure 2**

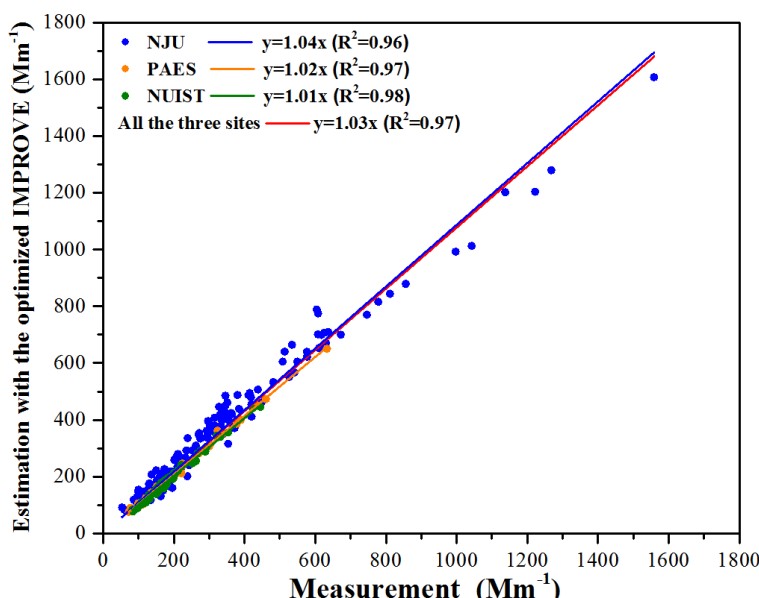







**Figure 3**

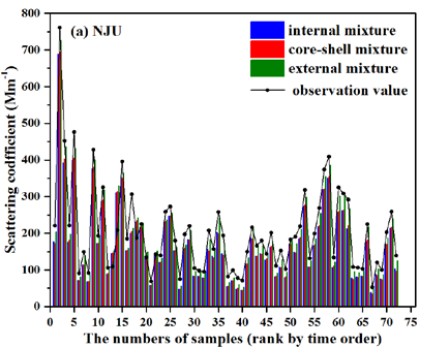
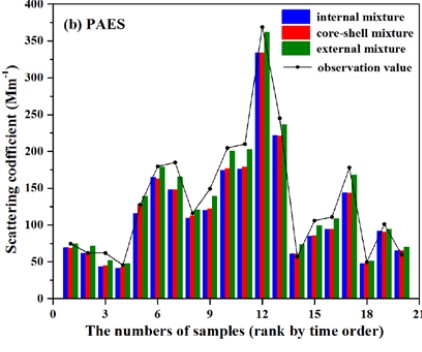

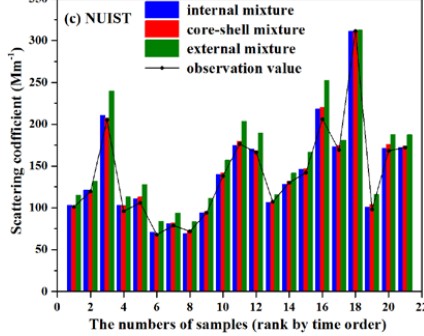







**Figure 4**

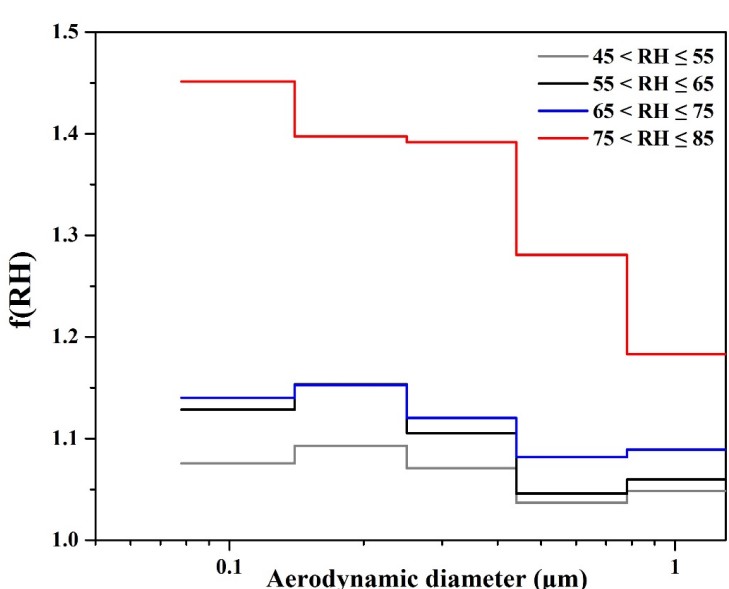








**Figure 5**

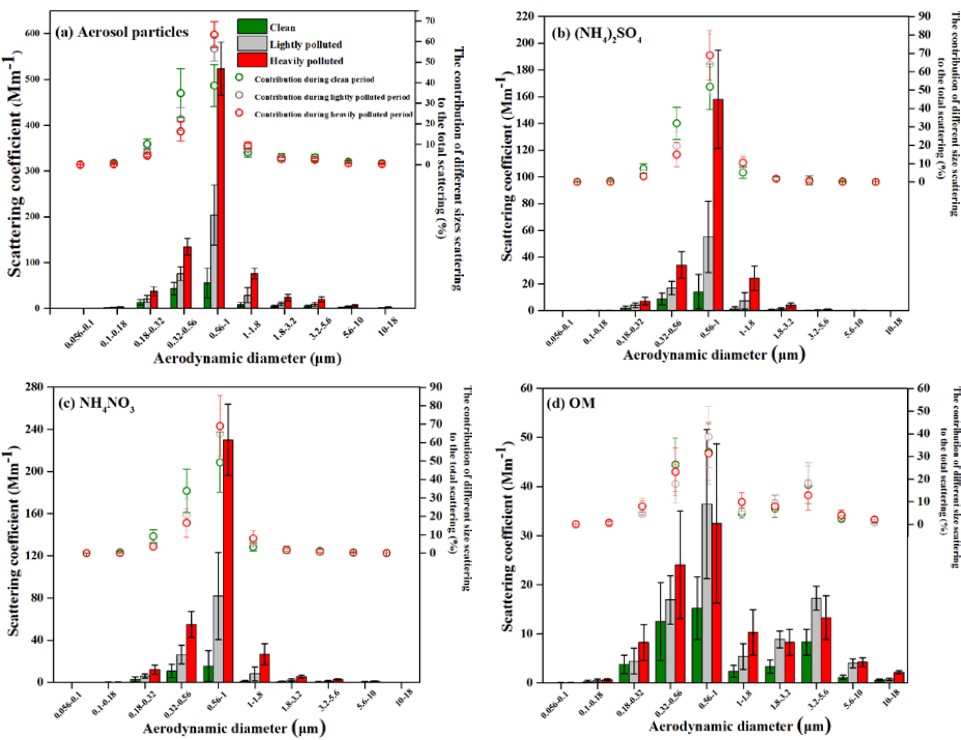







**Figure 6**

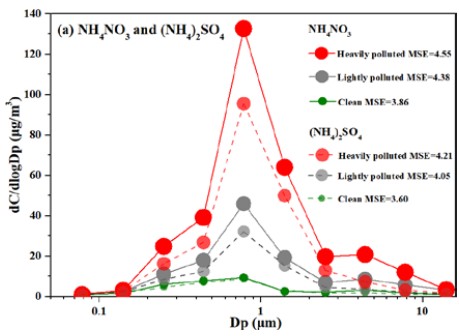
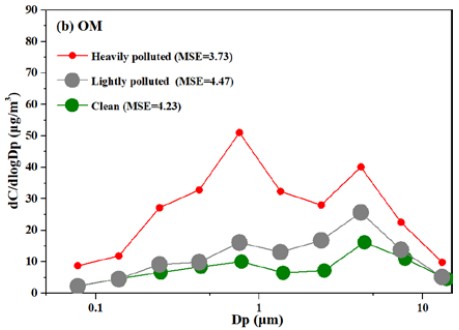







**Figure 7**

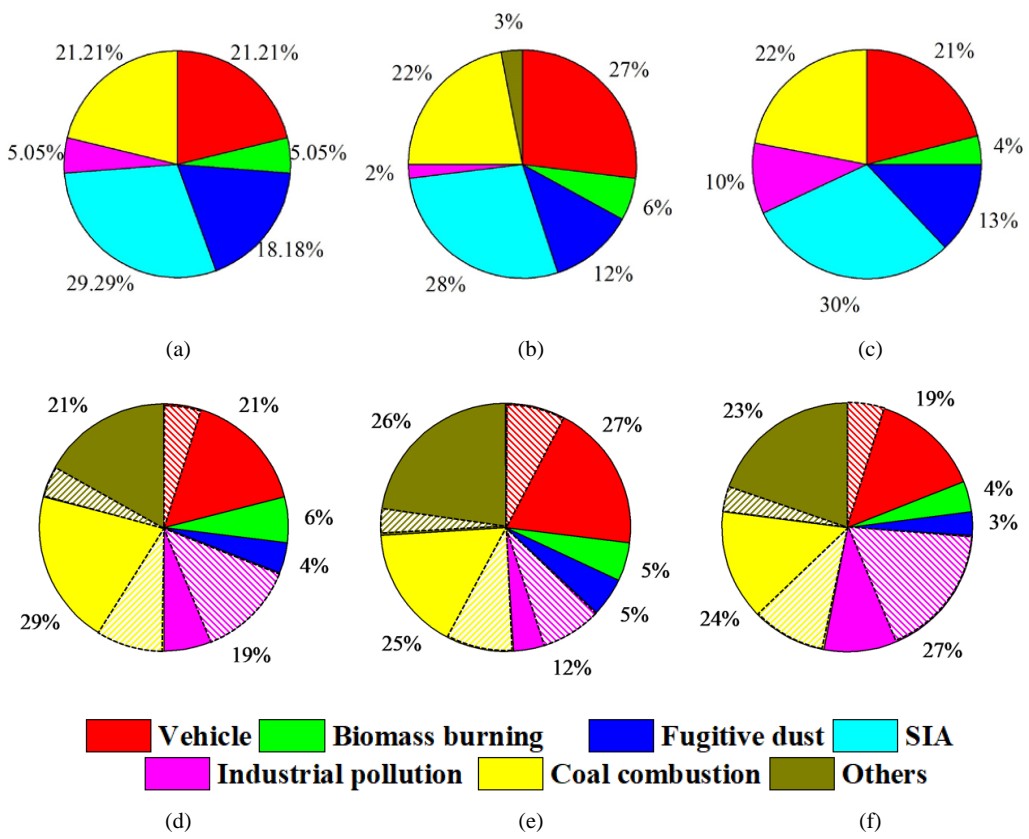




**Figure 8**

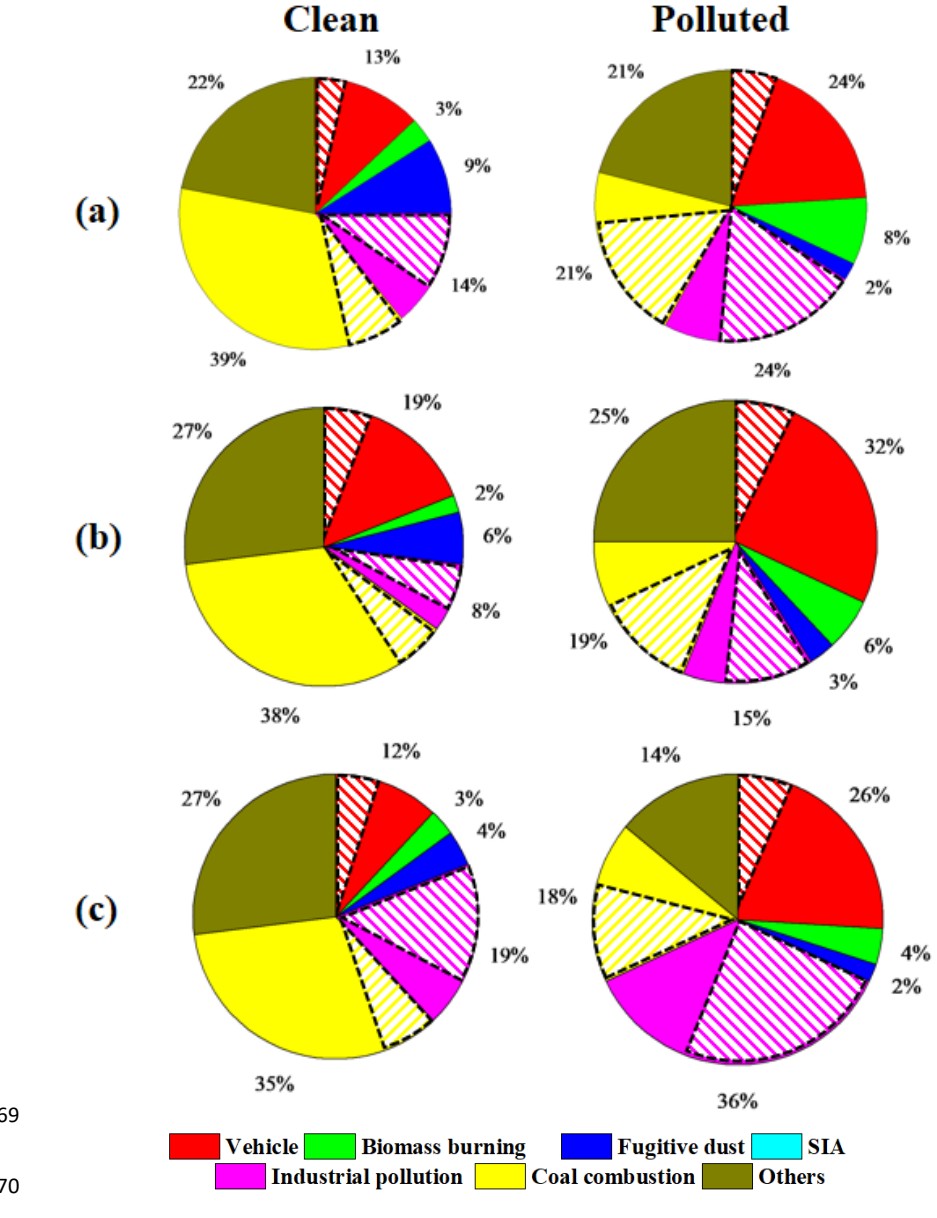

