# Peer review of "Characterization and source apportionment of aerosol light"

_Atmospheric Chemistry and Physics, 2020_

## Referee Comment (RC1) · Anonymous Referee #1 · 26 May 2020

The paper by Chen et al. systematically investigates the characteristics and sources of aerosol light scattering through measurements at three different functional sites in a typical polluted city in the Yangtze River Delta, China. Aerosol scattering is important for both visibility degradation and air pollution, and is also complex due to aerosol chemical composition and hygroscopic growth. In this study, the US IMPROVE formula for aerosol scattering calculation was optimized using online and offline measurements at different functional sites in Nanjing with complicated sources of air pollutants. The influence of aerosol size distributions and pollution levels on the aerosol scattering was quantitatively evaluated based on a comprehensive analysis of the size-specific chemical compositions of particles at various sites. In general, this manuscript is well

organized and easy to follow. I would recommend its acceptance after some necessary corrections suggested as follows:

1. Line 87: "NH4NO3 and (NH4)2SO4" need to be defined at their first mention in the manuscript. The manuscript has similar problems with other chemical species as well. Please go through the manuscript and change all of them.

2. Line 149: What is the mass fraction of the methanol soluble organic carbon in the total organic carbon mass? Did you try the water extraction?

3. Line 186: In the process of formula optimization, why did the authors subtract the scattering coefficients by sea salt, soil dust and coarse particles from the measured scattering coefficient? Does it mean that the light scattering of those species has little impact on the optimization of IMPROVE formula?

4. Line 201: Mie theory is very sensitive to the size distribution of aerosol chemical species. However, the size distribution data obtained from a high-flow MOUDI impactor can usually be influenced by the particle bounce. This is particularly concerned in case where filters, instead of metal foils with grease coating, are used as the substrate. I suggest the authors make an uncertainty evaluation upon the size distribution measurement in this study.

5. In Section 3.2, the US IMPROVE algorithm was optimized only within one city in the Yangtze River Delta with good performance. How did the authors consider the application of the optimized formula in typical regions such as cities in Beijing-Tianjin-Hebei or Pearl River Delta? Some discussions are recommended here.

6. Line 352: The study did not mention if the scattering coefficients used for the US IMPROVE estimation at the three sites were measured by CAPS or nephelometer? According to Section 2.3, the scattering coefficients at PAES and NUIST were measured by two integrating nephelometers. Need some clarification on this issue.

7. Line 447: Due to the varied chemical properties of particles in different regions, the

growth factors of particles (GF) can be different, and it would bring some uncertainty to the calculation of scattering coefficient in Section 3.3. It is recommended to measure and apply the local GF values in this work.

8. In Section 3.4, there was no clear description whether the scattering coefficients were estimated based on the assumption of dry or ambient conditions.

9. In Section 3.5, the assumption that the secondary components were proportional to the emissions of their precursors is subject to great uncertainty, as noted by the authors. Please be more specific on how to get better results with improved measurement or modeling methods.

Some minor comments:

Line 31: Define "IMPROVE" on first usage.

Line 32: "OC" should not be abbreviated when it is mentioned for the first time.

Line 160: What is the wavelength of the integrating nephelometer at the three sites used?

Line 246: The operational symbol was missing in Eq. (3).

Line 522: "Mien theory" should be "Mie theory".

Line 562: "PME" should be "PMF"

Line 970: SIA in the legend did not exist in Figure 8.

Reference list: The format of references should be in accordance with the journal requirement.
* * *

---

## Referee Comment (RC2) · Anonymous Referee #2 · 27 May 2020

The current manuscript presents a comprehensive study of the influential factors and source apportionment of aerosol light scattering at three sites in Nanjing, representative for suburban (NJU), urban (PAES) and industrial areas (NUIST) respectively. The data obtained in this work show interesting details on the linkage between chemistry composition and light scattering of aerosols, and help better understanding the effects of various sources on visibility degradation at the city scale. Overall I think the work provides reasonable analysis and the paper is clearly written. Before it can be published in Atmospheric Chemistry and Physics, however, I have some concerns that should be further addressed, and minor revisions are accordingly suggested as below. 1. In line 130, QA/QC procedures of aerosol sampling process are missed in this

manuscript, which are important for a scientific paper presenting the first-hand data. For example, the MOUDI sampler could be blocked during heavy pollution conditions, and the collected samples might not be evenly distributed. This phenomenon would affect the chemical analysis, particularly for OC and EC (choice of spots). How did the authors treat such kind of problems or estimate the uncertainty from sampling? 2. In Line 136, were field blanks obtained during the sampling campaigns? And, why were the sampling periods different at the three sites? Similarly, in Line 128, why was the sampling size at NJU larger than another two sites? The sampling strategy should be described more. 3. In Line 145, Sunset analyzer was able to measure thermal EC and OC, and optical EC and OC. The author should clarify it carefully in the paper. Why choose them for the analysis? 4. In Line 183, what software did authors use to run the multiple linear regression? If this model has been developed or used in other studies, the references should be provided. 5. In Line 190, considering the light absorption of methanol soluble organic carbon (MSOC), the optimization of the US IMPROVE algorithm is quite interesting. How did the authors estimate the MSOC concentrations in fine and large size modes? 6. In Line 190, why did the authors only include those eight variables in this equation? How about other species like coarse particles, sea salt, and soil dust? 7. In Line 214, the authors need to explain the special reason why they applied PMF model in their analysis. 8. In Line 292, the author stated that "the sum of $NO_3^-$, $SO_4^{2-}$ and OC for the heavily polluted period was 10.7 and 2.9 times greater than those for the lightly polluted and clean periods". Something seems wrong here. In which period was the concentration higher, lightly polluted or clean period? From Table 1, moreover, it seems that SNA was elevated more than OC in the heavily polluted period compared to the clean days. Any reason for this difference? 9. The authors did not clearly explain the data of which site were used in Section 3.4. If it is based on the data of the three sites, the assumption in Section 3.4 that chemical particles were externally mixed will cause large uncertainty in the calculation of scattering coefficient at NUIST because the internal mixt was an important particle mixing state at NUIST (Figure 3c). 10. In Line 534, in general, results generated from PMF model could be

questionable if less than 100 samples were used in the model. How did the authors consider this problem?

---

## Author Comment (AC1) · 8 Jul 2020

We thank very much for the valuable comments and suggestions from the reviewer, which help us improve our manuscript significantly. The comments were carefully considered and revisions have been made in response to suggestions. Following is our point-by-point responses to the comments and corresponding revisions.

0. The paper by Chen et al. systematically investigates the characteristics and sources of aerosol light scattering through measurements at three different functional sites in a typical polluted city in the Yangtze River Delta, China. Aerosol scattering is important for both visibility degradation and air pollution, and is also complex due to aerosol

chemical composition and hygroscopic growth. In this study, the US IMPROVE formula for aerosol scattering calculation was optimized using online and offline measurements at different functional sites in Nanjing with complicated sources of air pollutants. The influence of aerosol size distributions and pollution levels on the aerosol scattering was quantitatively evaluated based on a comprehensive analysis of the size-specific chemical compositions of particles at various sites. In general, this manuscript is well organized and easy to follow. I would recommend its acceptance after some necessary corrections suggested as follows:

Response and revisions:

We appreciate the reviewer's positive remarks on the importance of the work.

1. Line 87: "NH4NO3 and (NH4)2SO4" need to be defined at their first mention in the manuscript. The manuscript has similar problems with other chemical species as well. Please go through the manuscript and change all of them.

Response and revisions:

We thank the reviewer's reminder. As suggested, the two species were defined at their first appearance (lines 88 in the revised manuscript). We have also checked through the manuscript and revised all other items that need to be defined.

2. Line 149: What is the mass fraction of the methanol soluble organic carbon in the total organic carbon mass? Did you try the water extraction?

Response and revisions:

We thank the reviewer's comment. We explored the relationship between the total organic carbon (OC) and methanol soluble organic carbon (MSOC) concentrations in this study. The average MSOC was $8.23 \pm 4.84$ $\mu$g/m3 and accounted for 88% of the total OC mass in all samples. This result was similar to the fraction of 85% estimated by Cheng et al. (2016). Considering that a large fraction of brown carbon (BrC) absorption comes from OC insoluble in water, water extraction (WSOC) method was thus not

applied in the current study. We have added the discussion in lines 174-181 in the revised manuscript, and provided a new Figure S4 in the revised supplement, illustrating the relationship between the total OC and MSOC concentrations

3. Line 186: In the process of formula optimization, why did the authors subtract the scattering coefficients by sea salt, soil dust and coarse particles from the measured scattering coefficient? Does it mean that the light scattering of those species has little impact on the optimization of IMPROVE formula?

Response and revisions:

We thank the reviewer's comment. We calculate the ratios of the collective scattering coefficients of the sea salt and soil dust to the total PM2.5 scattering at the three sites. The ratios were 0.083, 0.093 and 0.081 at NJU, PAES, and NUIST, respectively, i.e., the scattering coefficients by sea salt and soil dust accounted for less than 10% of the total PM2.5 light scattering. Therefore the impact of the two species on the optimization of IMPROVE algorithm should be limited. In order to be concise in the algorithm optimization and to ensure the stability of the multiple linear regression, the independent variables contained $(NH_4)_2SO_4$, $NH_4NO_3$ and OM in the optimized formula, and the light scattering of sea salt and soil dust was subtracted from the measured scattering coefficient of PM2.5. We have clarified the methodology in lines 217-220 in the revised manuscript, and have added the discussion in lines 459-465 in the revised manuscript. A new Figure S10 has also been provided in the revised supplement, illustrating the ratios of the collective scattering coefficients of the sea salt and soil dust to the total PM2.5 scattering at the three sites.

4. Line 201: Mie theory is very sensitive to the size distribution of aerosol chemical species. However, the size distribution data obtained from a high-flow MOUDI impactor can usually be influenced by the particle bounce. This is particularly concerned in case where filters, instead of metal foils with grease coating, are used as the substrate. I suggest the authors make an uncertainty evaluation upon the size distribution

measurement in this study.

Response and revisions:

We thank the reviewer's comment. Although application of metal foils with grease coating could avoid the particle bounce, it might change the result of chemical species measurement. More, the metal foils substrate cannot meet the requirement of carbonaceous aerosol analysis, due to its special heating up program. In this study, therefore, we selected teflon filter for ion and element analysis, and quart fiber filter for carbonaceous aerosol analysis. Teflon filter membrane was generally applied for size-resolved particles sampling by MOUDI with excellent results (Contini et al., 2014; Gao et al., 2016; Guo et al., 2010). Taking NJU as an example, excellent agreement was found between the mass concentrations of PM1.8 collected with quartz fiber in MOUDI impactor and PM2.5 collected with TH-150 samplers. Therefore, the effect of particle bounce was expected to limited in this study. We have added the information in lines 157-161 in the revised manuscript, and provided a new Figure S3 (Figure R3 here) in the revised supplement, illustrating the correlation between the mass concentrations of PM1.8 collected with MOUDI impactor and PM2.5 collected with TH-150C sampler at NJU.

5. In Section 3.2, the US IMPROVE algorithm was optimized only within one city in the Yangtze River Delta with good performance. How did the authors consider the application of the optimized formula in typical regions such as cities in Beijing-Tianjin-Hebei or Pearl River Delta? Some discussions are recommended here.

Response and revisions:

We thank the reviewer's comment. In this study, the optimized IMPROVE formula was obtained based on the measured ambient concentrations of aerosol chemical species at three different functional sites in Nanjing, a typical polluted city in the Yangtze River Delta. As the chemical composition of aerosol (particularly SNA) was the key factor affecting its light scattering, the optimized IMPROVE formula could be applied in nearby

cities with similar composition of aerosols in eastern China including Shanghai and Jinan, as we stated in lines 465-474 in the revised manuscript. Moreover, for other regions with rapidly developing economy and fast industrialization in China including Beijing-Tianjin-Hebei or Pearl River Delta regions, the current work provides methodology and data support for the studies of aerosol light scattering in cities with relatively serious particle pollution. Given the fast changes in emission control and aerosol pollution in those regions, more campaigns on aerosol optical and chemical properties are recommended to further evaluate and improve the applicability of the optimized IMPROVE algorithm. We have added the explanation in lines 474-481 in the revised manuscript.

6. Line 352: The study did not mention if the scattering coefficients used for the US IMPROVE estimation at the three sites were measured by CAPS or nephelometer? According to Section 2.3, the scattering coefficients at PAES and NUIST were measured by two integrating nephelometers. Need some clarification on this issue.

Response and revisions:

We thank the reviewer's reminder and sorry for the error. The scattering coefficients used for the evaluation and optimization of the IMPROVE algorithm at the three sites were all measured with nephelometers. The relevant texts have been revised in line 399 in the revised manuscript.

7. Line 447: Due to the varied chemical properties of particles in different regions, the growth factors of particles (GF) can be different, and it would bring some uncertainty to the calculation of scattering coefficient in Section 3.3. It is recommended to measure and apply the local GF values in this work.

Response and revisions:

We thank and agree the reviewer's comment. Due to the lack of suitable instrument like hygroscopicity tandem differential mobility analyzer (H-TDMA), we did not measure

the local GF values directly, and it is a limitation of this study. Instead, we collected the GF data from the existing local studies in Nanjing (Table S1 in the supplement), and applied them in estimation of ambient scattering coefficient by Mie theory. To check the uncertainty of this application, the estimates were compared with those calculated with the scattering hygroscopic growth factor (f(RH)), as shown in Figure S12 in the revised supplement (Figure S8 in the original submission). A good agreement was found between the two methods (R2=0.95), indicating the limited uncertainty from the GF values applied in this study.

8. In Section 3.4, there was no clear description whether the scattering coefficients were estimated based on the assumption of dry or ambient conditions.

Response and revisions:

We thank the reviewer's reminder. The estimated scattering coefficients in Section 3.4 were based on the assumption of ambient condition. The relevant text has been revised in line 543 in the revised manuscript.

9. In Section 3.5, the assumption that the secondary components were proportional to the emissions of their precursors is subject to great uncertainty, as noted by the authors. Please be more specific on how to get better results with improved measurement or modeling methods.

Response and revisions:

We thank the reviewer's comment. As we stated in the manuscript, there was substantial uncertainty in the methodology in which source apportionment of secondary aerosols depends on the magnitudes of precursor emissions. It is a limitation of the present study. To further improve the source apportionment results, some specific tracers of secondary aerosols like semi volatile and low volatile oxygen-containing organic aerosols can be firstly observed with advanced technology such as aerosol mass spectrometry (AMS), and the observation data can then be combined with receptor models

to quantify the source contribution of secondary aerosols. Besides, air quality model that integrates particle source apportionment technology (PSAT) is recommended to be applied to evaluate and confirm the performance of the source apportionment of secondary aerosol with the receptor model. We have added the explanation in lines 678-685 in the revised manuscript.

10. Some minor comments: Line 31: Define "IMPROVE" on first usage.

Response and revisions:

We thank the reviewer's reminder and the full name has been given in the revised manuscript.

Line 32: "OC" should not be abbreviated when it is mentioned for the first time.

Response and revisions:

We thank the reviewer's reminder and the full name has been given in the revised manuscript.

Line 160: What is the wavelength of the integrating nephelometer at the three sites used?

Response and revisions:

We thank the reviewer's reminder. The two nephelometers (Ecotech Pty Ltd, Australia, Model Aurora1000G) at NJU and PAES were operated at the wavelength of 520 nm. The integrating nephelometer (Model 3563, TSI, USA) used at NUIST can measure the light scattering at three visible wavelengths (450, 550 and 700 nm), and the scattering coefficient at the wavelength of 550 nm was adopted in this work. We have added the explanation in lines 190-193 in the revised manuscript.

Line 246: The operational symbol was missing in Eq. (3).

Response and revisions:

We thank the reviewer's reminder and it is corrected in the revised manuscript.

Line 522: "Mien theory" should be "Mie theory".

Response and revisions:

We are sorry for this mistake and thanks for the reminder. We have corrected it in the revised manuscript.

Line 562: "PME" should be "PMF".

Response and revisions:

We are sorry for this mistake and thanks for the reminder. We have corrected it in the revised manuscript.

Line 970: SIA in the legend did not exist in Figure 8.

Response and revisions:

We thank the reviewer's reminder and the SIA legend in Figure 8 has been removed.

Reference list: The format of references should be in accordance with the journal requirement.

Response and revisions:

We thank the reviewer's reminder. We have checked the format of references and made it consistent with the journal requirement.

References

Cheng, Y., He, K.B., Du, Z.Y., Engling, G., Liu, J.M., Ma, Y.L., Zheng, M., Weber, R.J.: The characteristics of brown carbon aerosol during winter in Beijing, Atmos. Environ., 127, 355-364, doi:10.1016/j.atmosenv.2015.12.035, 2016.

Contini, D., Cesari, D., Genga, A., Sicilianob, M., Ielpo, P., Guascito, M. R., Conte, M.: Source apportionment of size-segregated atmospheric particles based on the

major water-soluble components in Lecce (Italy), Sci. Total Environ., 472, 248-261, 10.1016/j.scitotenv.2013.10.127, 2014.

Gao, Y., Lee, S. C., Huang, Y., Chow, J. C., Watson, J. G.: Chemical characterization and source apportionment of size-resolved particles in Hong Kong sub-urban area, Atmos. Res., 170, 112-122, 10.1016/j.atmosres.2015.11.015, 2016.

Guo, S., Hu, M., Wang, Z. B., Slanina, J., and Zhao, Y. L.: Size resolved aerosol water-soluble ionic compositions in the summer of Beijing: implication of regional secondary formation, Atmos. Chem. Phys., 10, 947–959, doi:10.5194/acp-10-947-2010, 2010.

---

## Author Comment (AC2) · 8 Jul 2020

We thank very much for the valuable comments and suggestions from the reviewer, which help us improve our manuscript significantly. The comments were carefully considered and revisions have been made in response to suggestions. Following is our point-by-point responses to the comments and corresponding revisions.

0. The current manuscript presents a comprehensive study of the influential factors and source apportionment of aerosol light scattering at three sites in Nanjing, representative for suburban (NJU), urban (PAES) and industrial areas (NUIST) respectively. The data obtained in this work show interesting details on the linkage between chemistry composition and light scattering of aerosols, and help better understanding the effects of various sources on visibility degradation at the city scale. Overall I think the work provides reasonable analysis and the paper is clearly written. Before it can be published in Atmospheric Chemistry and Physics, however, I have some concerns that should be further addressed, and minor revisions are accordingly suggested as below.

Response and revisions:

We appreciate the reviewer's positive remarks on the importance of the work.

1. In line 130, QA/QC procedures of aerosol sampling process are missed in this manuscript, which are important for a scientific paper presenting the first-hand data. For example, the MOUDI sampler could be blocked during heavy pollution conditions, and the collected samples might not be evenly distributed. This phenomenon would affect the chemical analysis, particularly for OC and EC (choice of spots). How did the authors treat such kind of problems or estimate the uncertainty from sampling?

Response and revisions:

We thank the reviewer's comment. To prevent the blocking by particles during sampling, the MOUDI samplers were first cleaned using an ultrasonic bath for 30 min before each sampling. In addition, the sampling flow rate was calibrated before each sampling and was also monitored with the flow meter during the whole sampling period. Those quality control measures assured that the MOUDI samplers were not blocked during the sampling period. Even for heavily polluted days with the PM1.8 concentration measured at 128  $\mu$ gćm-3, the particles sampled by MOUDI were evenly distributed. We have added the explanation in lines 138-144 in the revised manuscript, and added a new Figure S2 (Figure R4 here) in the revised supplement, illustrating the size-resolved particle filter samples collected on 25 Dec 2015 at NJU.

2. In Line 136, were field blanks obtained during the sampling campaigns? And, why were the sampling periods different at the three sites? Similarly, in Line 128, why was

ACPD
the sampling size at NJU larger than another two sites? The sampling strategy should be described more.

Response and revisions:

We thank the reviewer's comment. Yes we applied field blanks to correct the possible bias in the analysis of aerosol chemical species. Totally 19 sets of size-segregated blank filters (10, 4 and 5 for NJU, PAES and NUIST, respectively) and 35 daily blank PM2.5 filters (25, 6 and 9 for NJU, PAES and NUIST, respectively) were obtained at the three sites. All the blank filters were put in the samplers without inlet air flow for 24 h when the field campaigns finished. We have added the information in lines 152-157 in the revised manuscript.

Attributed to weather condition and aerosol sampler maintenance, the sampling periods for the three sites were different. Simultaneous samplings were conducted at the three sites from one week to ten days in each season from summer 2016 to winter 2016-2017. For the remaining time, two MOUDI samplers were applied to collect Teflon and quartz filter samples simultaneously at one of the three sites. As the Cavity Attenuated Phase Shift Albedo monitor (CAPS) was only installed at NJU and large amounts of data on aerosol optical and chemical information were needed to examine the influence of relative humidity on aerosol light scattering (Section 3.3), the sampling size at NJU was larger than another two sites. We have added the explanation in lines 144-152 in the revised manuscript.

3. In Line 145, Sunset analyzer was able to measure thermal EC and OC, and optical EC and OC. The author should clarify it carefully in the paper. Why choose them for the analysis?

Response and revisions:

We thank the reviewer's comment. The Sunset analyzer provides both thermal and optical concentrations for carbonaceous aerosols, and thermal EC and OC were used in Interactive comment

this study. The instrument estimates the optical EC by measuring the light attenuation (ATN). As ATN was determined not only by EC but also by brown carbon (BrC), the optical method may overestimate the EC and thus underestimate the OC (Cui et al., 2016; Massabò et al., 2016). Therefore, we applied the measured thermal EC and OC in this study. We added the explanation in lines 169-173 in the revised manuscript.

4. In Line 183, what software did authors use to run the multiple linear regression? If this model has been developed or used in other studies, the references should be provided.

Response and revisions:

SPSS 16.0 was used to conduct the multiple linear regressions. This information was added in line 216 in the revised manuscript and relative references were provided (Cheng et al., 2011; Tian et al., 2016).

5. In Line 190, considering the light absorption of methanol soluble organic carbon (MSOC), the optimization of the US IMPROVE algorithm is quite interesting. How did the authors estimate the MSOC concentrations in fine and large size modes?

Response and revisions:

The estimations of fine MSOC and large MSOC concentrations were the same as the calculations of ammonium nitrate (NH4NO3), ammonium sulfate ((NH4)2SO4) and organic carbon (OM), following the previous studies (Pitchford et al., 2007; Cheng et al., 2015). The concentration of large MSOC was estimated by dividing the total concentration of the component by 20  $\mu$ g/m3 (e.g., if the total MSOC concentration was 4  $\mu$ g/m3, the large mode concentration was calculated to be one-fifth of 4  $\mu$ g/m3 or 0.8  $\mu$ g/m3, leaving 3.2  $\mu$ g/m3 in the small mode). If the total MSOC concentration exceeds 20  $\mu$ g/m3, all of it was assumed to be in the large mode.

6. In Line 190, why did the authors only include those eight variables in this equation? How about other species like coarse particles, sea salt, and soil dust?

**ACPD**
Response and revisions:

We thank the reviewer's comment. In this study, the optimization of the US IMPROVE formula was for PM2.5, thus the optimization process did not include coarse particles. As indicated in Figure R2 (the response to Question 3 of reviewer 1), the scattering coefficients by sea salt and soil dust accounted for less than 10% of the total PM2.5 scattering, suggesting that the two species should have limited contribution to the total scattering coefficient. In order to be concise in the IMPROVE formula optimization and to ensure the stability of the multiple linear regression, therefore, only ammonium sulfate, ammonium nitrate and organic matter were used as independent variables in the regression. We have added the explanation in lines 459-465 in the revised manuscript.

7. In Line 214, the authors need to explain the special reason why they applied PMF model in their analysis.

Response and revisions:

We thank the reviewer's comment. The source apportionment technologies include emissions inventory, chemistry transport model and the receptor model, and the receptor model based on aerosol chemistry analysis has been widely used because it is not limited by the uncertainty of emission inventory or meteorology simulation. Principally the receptor models contain two categories, i.e., the models in which source profiles are needed, such as the chemical mass balance (CMB) method, and those in which source profiles are not needed, such as the positive matrix factorization (PMF) method (Yin et al., 2015). Due to lack of sufficient local measurements, it is generally difficult to build comprehensive and accurate source profiles of various aerosol components for individual cities in China. In this work, therefore, we applied the PMF model to exclude the uncertainty of source profiles which were commonly developed based on literatures or measurements across the country. We have added the explanation in lines 247-252 in the revised manuscript.
8. In Line 292, the author stated that "the sum of NO3-, SO42- and OC for the heavily polluted period was 10.7 and 2.9 times greater than those for the lightly polluted and clean periods". Something seems wrong here. In which period was the concentration higher, lightly polluted or clean period? From Table 1, moreover, it seems that SNA was elevated more than OC in the heavily polluted period compared to the clean days. Any reason for this difference?

Response and revisions:

We thank the reviewer's comment. We are sorry for the mistakes and corrected the text as "the sum of NO3-, SO42- and OC for the heavily polluted period was 2.9 and 10.7 times greater than those for the lightly polluted and clean periods" in line 328 in the revised manuscript.

It is correct that SNA concentration was elevated more than OC in the heavily polluted period compared to the clean days in this work. During heavily polluted episodes, enhancement of sulfate and nitrate levels could be more significant than organic matter because the high relative humidity and precursor emissions (i.e., SO2 and NOx) promoted the generation of SNA (Tian et al., 2014). During clean periods (commonly in summer), ammonium nitrate (NH4NO3) would dissociate to NH3 and HNO3 at high temperatures, while secondary organic carbon (SOC) concentration may be increased due to the high levels of O3 and solar radiation. Those factors caused the result that SNA was elevated more than OC in the heavily polluted period compared to the clean days. Similar result was found in Beijing, where the mass fraction of SNA in fine particles increased from 19% on non-haze days to 31% on haze days, while that of OM decreased from 38% on non-haze days to 31% on haze days (Tian et al., 2016). We have added the above explanation in lines 334-342 in the revised manuscript.

9. The authors did not clearly explain the data of which site were used in Section 3.4. If it is based on the data of the three sites, the assumption in Section 3.4 that chemical particles were externally mixed will cause large uncertainty in the calculation of scatter-

**ACPD**
ing coefficient at NUIST because the internal mixture was an important particle mixing state at NUIST (Figure 3c).

Response and revisions:

We thank the reviewer's comment. Assuming different aerosol species were externally mixed, the influences of size distribution and pollution levels on aerosol light scattering was analyzed based on the aerosol composition information at the three sites. Although internal mixing was identified as an important particle mixing state at NUIST, the Mie theory cannot simulate the scattering coefficients of individual aerosol chemical components based on the assumption of internal or core-shell mixing but external mixing (Cheng et al., 2015; Ding et al 2015). According to Figure 3c in the revised manuscript, the scattering coefficient estimated with the external mixing assumption was  $1.07\pm0.05$  and  $1.08\pm0.06$  times of those from the internal mixing state simulation and instrument measurement, respectively. Therefore, the overestimation was smaller than 10% by the aggregated scattering coefficient from those of individual chemical species with an assumption of external mixing at NUIST. We have added the explanation in lines 540-545 in the revised manuscript.

10. In Line 534, in general, results generated from PMF model could be questionable if less than 100 samples were used in the model. How did the authors consider this problem?

Response and revisions:

In this study, the PMF analysis was performed respectively at the three sites for the accumulation mode particles with the size bins from 0.18 to 1.8  $\mu$ m. The samples used in PMF model were 300, 100 and 124 at NJU, PAES and NUIST, respectively. We thus believe that the sample size was sufficient for the PMF analysis.

References

Cheng, S. H., Yang, L. X., Zhou, X. H., Xue, L. K., Gao, X. M., Zhou, Y., and Wang, W.
X.: Size-fractionated water-soluble ions, situ pH and water content in aerosol on hazy days and the influences on visibility impairment in Jinan, China, Atmos. Environ., 45, 4631–4640, doi:10.1016/j.atmosenv.2011.05.057, 2011.

Cheng, Z., Jiang, J., Chen, C., Gao, J., Wang, S., Watson, J.G., Wang, H., Deng, J., Wang, B., Zhou, M.: Estimation of aerosol mass scattering efficiencies under high mass loading: case study for the megacity of Shanghai, China, Environ. Sci. Technol., 49, 831-838, doi:10.1021/es504567q, 2015.

Cui, X., Wang, X., Yang, L., Chen, B., Chen, J., Andersson, A., Gustafsson, Ö.: Radiative absorption enhancement from coatings on black carbon aerosols, Sci. Total Environ., 551-552, 51-56, doi:10.1016/j.scitotenv.2016.02.026, 2016.

Ding, J., Han, S., Zhang, Y., Feng, Y., Wu, J., Shi, G., Wang, J.: Chemical characteristics of particles and light extinction effects in winter in Tianjin, Res. Environ. Sci., 28, 1353-1361, doi:10.13198/j.issn.1001-6929.2015.09.03, 2015.

Massabò, D., Caponi, L., Bove, M.C., Prati, P.: Brown carbon and thermal-optical analysis: A correction based on optical multi-wavelength apportionment of atmospheric aerosols, Atmos. Environ., 125, 119-125, doi:10.1016/j.atmosenv.2015.11.011, 2016.

Pitchford, M., Malm, W., Schichtel, B., Kumar, N., Lowenthal, D., Hand, J.: Revised algorithm for estimating light extinction from IMPROVE particle speciation data, J. Air Waste Manag. Assoc., 57, 1326-1336, doi:10.3155/1047-3289.57.11.1326, 2007.

Tian, S. L., Pan, Y. P., Liu, Z., Wen, T., and Wang, Y. S.: Size-resolved aerosol chemical analysis of extreme haze pollution events during early 2013 in urban Beijing, China, J. Hazard. Mater., 279, 452–460, doi:10.1016/j.jhazmat.2014.07.023, 2014.

Tian, S. L., Pan, Y. P., Wang, Y. S.: Size-resolved source apportionment of particulate matter in urban Beijing during haze and non-haze episodes, Atmos. Chem. Phys., 16, 1-19, doi:10.5194/acp-16-1-2016, 2016.

Yin, J., Cumberland, S. A., Harrison, R. M., Allan, J., Young, D. E., Williams, P. I., and
Coe, H.: Receptor modelling of fine particles in southern England using CMB including comparison with AMS-PMF factors, Atmos Chem Phys, 15, 2139-2158, 2015.

---

## Author Response (AR1)

**Main revisions and response to reviewers' comments**

Manuscript No.: acp-2020-176

Title: Characterization and source apportionment of aerosol light scattering in a typical polluted city in Yangtze River Delta, China

Authors: Dong Chen, Yu Zhao, Jie Zhang, Huan Yu, Xingna Yu

We thank very much for the valuable comments and suggestions from the two reviewers, which help us improve our manuscript significantly. The comments were carefully considered and revisions have been made in response to suggestions. Following is our point-by-point responses to the comments and corresponding revisions.

**Reviewer #1**

**0.** The paper by Chen et al. systematically investigates the characteristics and sources of aerosol light scattering through measurements at three different functional sites in a typical polluted city in the Yangtze River Delta, China. Aerosol scattering is important for both visibility degradation and air pollution, and is also complex due to aerosol chemical composition and hygroscopic growth. In this study, the US IMPROVE formula for aerosol scattering calculation was optimized using online and offline measurements at different functional sites in Nanjing with complicated sources of air pollutants. The influence of aerosol size distributions and pollution levels on the aerosol scattering was quantitatively evaluated based on a comprehensive analysis of the size-specific chemical compositions of particles at various sites. In general, this manuscript is well organized and easy to follow. I would recommend its acceptance after some necessary corrections suggested as follows:

**Response and revisions:**

We appreciate the reviewer's positive remarks on the importance of the work.

1. Line 87: "$NH_4NO_3$ and $(NH_4)_2SO_4$" need to be defined at their first mention in the manuscript. The manuscript has similar problems with other chemical species as well. Please go through the manuscript and change all of them.

**Response and revisions:**

We thank the reviewer's reminder. As suggested, the two species were defined at their first appearance (**lines 88 in the revised manuscript**). We have also checked through the manuscript and revised all other items that need to be defined.

2. Line 149: What is the mass fraction of the methanol soluble organic carbon in the total organic carbon mass? Did you try the water extraction?

**Response and revisions:**

We thank the reviewer's comment. Figure R1 illustrates the relationship between the total organic carbon (OC) and methanol soluble organic carbon (MSOC) concentrations in this study. The average MSOC was $8.23 \pm 4.84$ μg/m$^3$ and accounted for 88% of the total OC mass in all samples. This result was similar to the fraction of 85% estimated by Cheng et al. (2016). Considering that a large fraction of brown carbon (BrC) absorption comes from OC insoluble in water, water extraction (WSOC) method was thus not applied in the current study. We have added the discussion **in lines 174-181 in the revised manuscript**, and provided Figure R1 as **a new Figure S4 in the revised supplement**.

[Figure]

Figure R1. Correlation between MSOC and total OC for all the samples collected in this work. The red line indicates linear regression results with K as slope (intercept is set at zero).

3. Line 186: In the process of formula optimization, why did the authors subtract the scattering coefficients by sea salt, soil dust and coarse particles from the measured scattering coefficient? Does it mean that the light scattering of those species has little impact on the optimization of IMPROVE formula?

**Response and revisions:**

We thank the reviewer's comment. Figure R2 presents the ratios of the collective scattering coefficients of the sea salt and soil to the total $PM_{2.5}$ scattering at the three sites. The ratios were 0.083, 0.093 and 0.081 at NJU, PAES, and NUIST, respectively, i.e., the scattering coefficients by sea salt and soil dust accounted for less than 10% of the total $PM_{2.5}$ light scattering. Therefore the impact of the two species on the optimization of IMPROVE algorithm should be limited. In order to be concise in the algorithm optimization and to ensure the stability of the multiple linear regression, the independent variables contained $(NH_4)_2SO_4$, $NH_4NO_3$ and OM in the optimized formula, and the light scattering of sea salt and soil dust was subtracted from the measured scattering coefficient of $PM_{2.5}$. We have clarified the methodology **in lines 217-220 in the revised manuscript**, and have added the discussion **in lines 459-465 in the revised manuscript**. **A new Figure S10** (i.e., Figure R2) has also been provided in the revised supplement.

[Figure]

Figure R2 The collective proportions of sea salt and soil dust to the total $PM_{2.5}$ scattering coefficient at the three sites based on the IMPROVE2007 algorithm.

4. Line 201: Mie theory is very sensitive to the size distribution of aerosol chemical species. However, the size distribution data obtained from a high-flow MOUDI impactor can usually be influenced by the particle bounce. This is particularly concerned in case where filters, instead of metal foils with grease coating, are used as the substrate. I suggest the authors make an uncertainty evaluation upon the size distribution measurement in this study.

**Response and revisions:**

We thank the reviewer's comment. Although application of metal foils with grease coating could avoid the particle bounce, it might change the result of chemical species measurement. More, the metal foils substrate cannot meet the requirement of carbonaceous aerosol analysis, due to its special heating up program. In this study, therefore, we selected teflon filter for ion and element analysis, and quart fiber filter for carbonaceous aerosol analysis. Teflon filter membrane was generally applied for size-resolved particles sampling by MOUDI with excellent results (Contini et al., 2014; Gao et al., 2016; Guo et al., 2010). Taking NJU as an example, excellent agreement was found between the mass concentrations of $PM_{1.8}$ collected with quartz fiber in MOUDI impactor and $PM_{2.5}$ collected with TH-150 samplers, as shown in Figure R3. Therefore, the effect of particle bounce was expected to limited in this study. We have added the information **in lines 157-161 in the revised manuscript**, and provided **a new Figure S3** (Figure R3 here) in the revised supplement.

[Figure]

Figure R3 Correlation between the mass concentrations of $PM_{1.8}$ collected with MOUDI impactor and $PM_{2.5}$ collected with TH-150C sampler at NJU.

5. In Section 3.2, the US IMPROVE algorithm was optimized only within one city in the Yangtze River Delta with good performance. How did the authors consider the application of the optimized formula in typical regions such as cities in Beijing-Tianjin-Hebei or Pearl River Delta? Some discussions are recommended here.

**Response and revisions:**

We thank the reviewer's comment. In this study, the optimized IMPROVE formula was obtained based on the measured ambient concentrations of aerosol chemical species at three different functional sites in Nanjing, a typical polluted city in the Yangtze River Delta. As the chemical composition of aerosol (particularly SNA) was the key factor affecting its light scattering, the optimized IMPROVE formula could be applied in nearby cities with similar composition of aerosols in eastern China including Shanghai and Jinan, as we stated **in lines 465-474 in the revised manuscript.** Moreover, for other regions with rapidly developing economy and fast industrialization in China including Beijing-Tianjin-Hebei or Pearl River Delta regions, the current work provides methodology and data support for the studies of aerosol light scattering in cities with relatively serious particle pollution. Given the fast changes in emission control and aerosol pollution in those regions, more campaigns on aerosol optical and chemical properties are recommended to further evaluate and improve the applicability of the optimized IMPROVE algorithm. We have added the explanation **in lines 474-481 in the revised manuscript**.

6. Line 352: The study did not mention if the scattering coefficients used for the US IMPROVE estimation at the three sites were measured by CAPS or nephelometer? According to Section 2.3, the scattering coefficients at PAES and NUIST were measured by two integrating nephelometers. Need some clarification on this issue.

**Response and revisions:**

We thank the reviewer's reminder and sorry for the error. The scattering coefficients used for the evaluation and optimization of the IMPROVE algorithm at the three sites were all measured with nephelometers. The relevant texts have been revised **in line 399 in the revised manuscript**.

7. Line 447: Due to the varied chemical properties of particles in different regions, the growth factors of particles (GF) can be different, and it would bring some uncertainty to the calculation of scattering coefficient in Section 3.3. It is recommended to measure and apply the local GF values in this work.

**Response and revisions:**

We thank and agree the reviewer's comment. Due to the lack of suitable instrument like hygroscopicity tandem differential mobility analyzer (H-TDMA), we did not measure the local GF values directly, and it is a limitation of this study. Instead, we collected the GF data from the existing local studies in Nanjing (Table S1 in the supplement), and applied them in estimation of ambient scattering coefficient by Mie theory. To check the uncertainty of this application, the estimates were compared with those calculated with the scattering hygroscopic growth factor (f(RH)), as shown in Figure S12 in the revised supplement (Figure S8 in the original submission). A good agreement was found between the two methods ($R^2$=0.95), indicating the limited uncertainty from the GF values applied in this study.

8. In Section 3.4, there was no clear description whether the scattering coefficients were estimated based on the assumption of dry or ambient conditions.

**Response and revisions:**

We thank the reviewer's reminder. The estimated scattering coefficients in Section 3.4 were based on the assumption of ambient condition. The relevant text has been revised **in line 543 in the revised manuscript**.

9. In Section 3.5, the assumption that the secondary components were proportional to the emissions of their precursors is subject to great uncertainty, as noted by the authors. Please be more specific on how to get better results with improved measurement or modeling methods.

**Response and revisions:**

We thank the reviewer's comment. As we stated in the manuscript, there was substantial uncertainty in the methodology in which source apportionment of secondary aerosols depends on the magnitudes of precursor emissions. It is a limitation of the present study. To further improve the source apportionment results, some specific tracers of secondary aerosols like semi volatile and low volatile oxygen-containing organic aerosols can be firstly observed with advanced technology such as aerosol mass spectrometry (AMS), and the observation data can then be combined with receptor models to quantify the source contribution of secondary aerosols. Besides, air quality model that integrates particle source apportionment technology (PSAT) is recommended to be applied to evaluate and confirm the performance of the source apportionment of secondary aerosol with the receptor model. We have added the explanation **in lines 678-685 in the revised manuscript**.

10. Some minor comments:

Line 31: Define "IMPROVE" on first usage.

**Response and revisions:**

We thank the reviewer's reminder and the full name has been given in the revised manuscript.

Line 32: "OC" should not be abbreviated when it is mentioned for the first time.

**Response and revisions:**

We thank the reviewer's reminder and the full name has been given in the revised manuscript.

Line 160: What is the wavelength of the integrating nephelometer at the three sites used?

**Response and revisions:**

We thank the reviewer's reminder. The two nephelometers (Ecotech Pty Ltd, Australia, Model Aurora1000G) at NJU and PAES were operated at the wavelength of 520 nm. The integrating nephelometer (Model 3563, TSI, USA) used at NUIST can measure the light scattering at three visible wavelengths (450, 550 and 700 nm), and the scattering coefficient at the wavelength of 550 nm was adopted in this work. We have added the explanation **in lines 190-193 in the revised manuscript**.

Line 246: The operational symbol was missing in Eq. (3).

**Response and revisions:**

We thank the reviewer's reminder and it is corrected in the revised manuscript.

Line 522: "Mien theory" should be "Mie theory".

**Response and revisions:**

We are sorry for this mistake and thanks for the reminder. We have corrected it in the revised manuscript.

Line 562: "PME" should be "PMF".

**Response and revisions:**

We are sorry for this mistake and thanks for the reminder. We have corrected it in the revised manuscript.

Line 970: SIA in the legend did not exist in Figure 8.

**Response and revisions:**

We thank the reviewer's reminder and the SIA legend in Figure 8 has been removed.

Reference list: The format of references should be in accordance with the journal requirement.

**Response and revisions:**

We thank the reviewer's reminder. We have checked the format of references and made it consistent with the journal requirement.

**Response and revisions:**

We thank the reviewer's comment. To prevent the blocking by particles during sampling, the MOUDI samplers were first cleaned using an ultrasonic bath for 30 min before each sampling. In addition, the sampling flow rate was calibrated before each sampling and was also monitored with the flow meter during the whole sampling period. Those quality control measures assured that the MOUDI samplers were not blocked during the sampling period. Even for heavily polluted days with the $PM_{1.8}$ concentration measured at 128 μg·m$^{-3}$, the particles sampled by MOUDI were evenly distributed, as shown in Figure R4. We have added the explanation **in lines 138-144 in the revised manuscript**, and added **a new Figure S2** (Figure R4 here) in the revised supplement.

[Figure]

| 5.6-10 μm | 3.2-5.6 μm | 1.8-3.2 μm | 1.0-1.8 μm | |
|---|---|---|---|---|
| 0.56-1.0 μm | 0.32-0.56 μm | 0.18-0.32 μm | 0.10-0.18 μm | 0.056-0.10 μm |

Figure R4 A set of size-resolved particle filter samples on 25 Dec 2015 at NJU. The measured $PM_{1.8}$ and $PM_{2.5}$ were 128 and 141μg/m$^3$, respectively.

2. In Line 136, were field blanks obtained during the sampling campaigns? And, why were the sampling periods different at the three sites? Similarly, in Line 128, why was the sampling size at NJU larger than another two sites? The sampling strategy should be described more.

**Response and revisions:**

We thank the reviewer's comment. Yes we applied field blanks to correct the possible bias in the analysis of aerosol chemical species. Totally 19 sets of size-segregated blank filters (10, 4 and 5 for NJU, PAES and NUIST, respectively) and 35 daily blank $PM_{2.5}$ filters (25, 6 and 9 for NJU, PAES and NUIST, respectively) were obtained at the three sites. All the blank filters were put in the samplers without inlet air flow for 24 h when the field campaigns finished. We have added the information **in lines 152-157 in the revised manuscript**.

Attributed to weather condition and aerosol sampler maintenance, the sampling periods for the three sites were different. Simultaneous samplings were conducted at the three sites from one week to ten days in each season from summer 2016 to winter 2016-2017. For the remaining time, two MOUDI samplers were applied to collect Teflon and quartz filter samples simultaneously at one of the three sites. As the Cavity Attenuated Phase Shift Albedo monitor (CAPS) was only installed at NJU and large amounts of data on aerosol optical and chemical information were needed to examine the influence of relative humidity on aerosol light scattering (Section 3.3), the sampling size at NJU was larger than another two sites. We have added the explanation **in lines 144-152 in the revised manuscript**.

3. In Line 145, Sunset analyzer was able to measure thermal EC and OC, and optical EC and OC. The author should clarify it carefully in the paper. Why choose them for the analysis?

**Response and revisions:**

We thank the reviewer's comment. The Sunset analyzer provides both thermal and optical concentrations for carbonaceous aerosols, and thermal EC and OC were used in this study. The instrument estimates the optical EC by measuring the light attenuation (ATN). As ATN was determined not only by EC but also by brown carbon (BrC), the optical method may overestimate the EC and thus underestimate the OC (Cui et al., 2016; Massabò et al., 2016). Therefore, we applied the measured thermal EC and OC in this study. We added the explanation **in lines 169-173 in the revised manuscript**.

4. In Line 183, what software did authors use to run the multiple linear regression? If this model has been developed or used in other studies, the references should be provided.

**Response and revisions:**

SPSS 16.0 was used to conduct the multiple linear regressions. This information was added **in line 216 in the revised manuscript** and relative references were provided (Cheng et al., 2011; Tian et al., 2016).

5. In Line 190, considering the light absorption of methanol soluble organic carbon (MSOC), the optimization of the US IMPROVE algorithm is quite interesting. How did the authors estimate the MSOC concentrations in fine and large size modes?

**Response and revisions:**

The estimations of fine MSOC and large MSOC concentrations were the same as the calculations of ammonium nitrate ($NH_4NO_3$), ammonium sulfate (($NH_4$)$_2SO_4$) and organic carbon (OM), following the previous studies (Pitchford et al., 2007; Cheng et al., 2015). The concentration of large MSOC was estimated by dividing the total concentration of the component by 20 μg/m$^3$ (e.g., if the total MSOC concentration was 4 μg/m$^3$, the large mode concentration was calculated to be one-fifth of 4 μg/m$^3$ or 0.8 μg/m$^3$, leaving 3.2 μg/m$^3$ in the small mode). If the total MSOC concentration exceeds 20 μg/m$^3$, all of it was assumed to be in the large mode.

6. In Line 190, why did the authors only include those eight variables in this equation? How about other species like coarse particles, sea salt, and soil dust?

**Response and revisions:**

We thank the reviewer's comment. In this study, the optimization of the US IMPROVE formula was for $PM_{2.5}$, thus the optimization process did not include coarse particles. As indicated in Figure R2 (the response to Question 3 of reviewer 1), the scattering coefficients by sea salt and soil dust accounted for less than 10% of the total $PM_{2.5}$ scattering, suggesting that the two species should have limited contribution to the total scattering coefficient. In order to be concise in the IMPROVE formula optimization and to ensure the stability of the multiple linear regression, therefore, only ammonium sulfate, ammonium nitrate and organic matter were used as independent variables in the regression. We have added the explanation **in lines 459-465 in the revised manuscript**.

7. In Line 214, the authors need to explain the special reason why they applied PMF model in their analysis.

**Response and revisions:**

We thank the reviewer's comment. The source apportionment technologies include emissions inventory, chemistry transport model and the receptor model, and the receptor model based on aerosol chemistry analysis has been widely used because it is not limited by the uncertainty of emission inventory or meteorology simulation. Principally the receptor models contain two categories, i.e., the models in which source profiles are needed, such as the chemical mass balance (CMB) method, and those in which source profiles are not needed, such as the positive matrix factorization (PMF) method (Yin et al., 2015). Due to lack of sufficient local measurements, it is generally difficult to build comprehensive and accurate source profiles of various aerosol components for individual cities in China. In this work, therefore, we applied the PMF model to exclude the uncertainty of source profiles which were commonly developed based on literatures or measurements across the country. We have added the explanation **in lines 247-252 in the revised manuscript**.

8. In Line 292, the author stated that "the sum of $NO_3^-$, $SO_4^{2-}$ and OC for the heavily polluted period was 10.7 and 2.9 times greater than those for the lightly polluted and clean periods". Something seems wrong here. In which period was the concentration higher, lightly polluted or clean period? From Table 1, moreover, it seems that SNA was elevated more than OC in the heavily polluted period compared to the clean days. Any reason for this difference?

**Response and revisions:**

We thank the reviewer's comment. We are sorry for the mistakes and corrected the text as "the sum of $NO_3^-$, $SO_4^{2-}$ and OC for the heavily polluted period was 2.9 and 10.7 times greater than those for the lightly polluted and clean periods" **in line 328 in the revised manuscript**.

It is correct that SNA concentration was elevated more than OC in the heavily polluted period compared to the clean days in this work. During heavily polluted episodes, enhancement of sulfate and nitrate levels could be more significant than organic matter because the high relative humidity and precursor emissions (i.e., $SO_2$ and $NO_x$) promoted the generation of SNA (Tian et al., 2014). During clean periods (commonly in summer), ammonium nitrate ($NH_4NO_3$) would dissociate to $NH_3$ and $HNO_3$ at high temperatures, while secondary organic carbon (SOC) concentration may be increased due to the high levels of $O_3$ and solar radiation. Those factors caused the result that SNA was elevated more than OC in the heavily polluted period compared to the clean days. Similar result was found in Beijing, where the mass fraction of SNA in fine particles increased from 19% on non-haze days to 31% on haze days, while that of OM decreased from 38% on non-haze days to 31% on haze days (Tian et al., 2016). We have added the above explanation **in lines 334-342 in the revised manuscript**.

9. The authors did not clearly explain the data of which site were used in Section 3.4. If it is based on the data of the three sites, the assumption in Section 3.4 that chemical particles were externally mixed will cause large uncertainty in the calculation of scattering coefficient at NUIST because the internal mixture was an important particle mixing state at NUIST (Figure 3c).

**Response and revisions:**

We thank the reviewer's comment. Assuming different aerosol species were externally mixed, the influences of size distribution and pollution levels on aerosol light scattering was analyzed based on the aerosol composition information at the three sites. Although internal mixing was identified as an important particle mixing state at NUIST, the Mie theory cannot simulate the scattering coefficients of individual aerosol chemical components based on the assumption of internal or core-shell mixing but external mixing (Cheng et al., 2015; Ding et al 2015). According to Figure 3c in the revised manuscript, the scattering coefficient estimated with the external mixing assumption was 1.07±0.05 and 1.08±0.06 times of those from the internal mixing state simulation and instrument measurement, respectively. Therefore, the overestimation was smaller than 10% by the aggregated scattering coefficient from those of individual chemical species with an assumption of external mixing at NUIST.

We have added the explanation **in lines 540-545 in the revised manuscript.**

10. In Line 534, in general, results generated from PMF model could be questionable if less than 100 samples were used in the model. How did the authors consider this problem?

**Response and revisions:**

In this study, the PMF analysis was performed respectively at the three sites for the accumulation mode particles with the size bins from 0.18 to 1.8 μm. The samples used in PMF model were 300, 100 and 124 at NJU, PAES and NUIST, respectively. We thus believe that the sample size was sufficient for the PMF analysis.

$$
\begin{aligned}
b_{sca} = {} & a \times f_S(RH)[Small\ (NH_4)_2SO_4] + b \times f_L(RH)[Large\ (NH_4)_2SO_4] \\
& + c \times f_S(RH)[Small\ NH_4NO_3] + d \times f_L(RH)[Large\ NH_4NO_3] \\
& + e \times \big([Small\ OM] - m \times [Small\ MSOC]\big) \\
& + f \times \big([Large\ OM] - n \times [Large\ MSOC]\big)

[revised manuscript text omitted]

**scattering at the three sites for the clean and polluted periods (%).**

| Air quality level | Sources | NJU | | PAES | | NUIST | |
|---|---|---|---|---|---|---|---|
| | | SIA | SOA | SIA | SOA | SIA | SOA |
| Clean | Coal combustion | 6.6 | 0.8 | 6.5 | 1.1 | 7.5 | 1.3 |
| | Industrial plants | 5.8 | 3.6 | 4.2 | 1.5 | 8.2 | 6.3 |
| | Vehicles | 2.1 | 1.0 | 6.1 | 1.5 | 4.2 | 1.1 |
| | Total | 19.9 | | 20.9 | | 28.6 | |
| Polluted | Coal combustion | 12.4 | 1.6 | 8.8 | 2.3 | 10.2 | 2.2 |
| | Industrial plants | 10.2 | 5.8 | 7.8 | 3 | 12.6 | 9.9 |
| | Vehicles | 5.0 | 1.7 | 7.9 | 2.6 | 5.2 | 1.6 |
| | Total | 36.7 | | 32.4 | | 41.7 | |

**Figure 1**

[Figure]

**Figure 2**

[Figure]

**Figure 3**

[Figure]

**Figure 4**

[Figure]

**Figure 5**

[Figure]

**Figure 6**

[Figure]

[Figure]

**Figure 7**

[Figure]

(a)          (b)          (c)

(d)          (e)          (f)

| ■ Vehicle | ■ Biomass burning | ■ Fugitive dust | ■ SIA |
| ■ Industrial pollution | ■ Coal combustion | ■ Others |

**Figure 8**

[Figure]